# Adaptively Robust Resettable Streaming

**Edith Cohen** [1,2]  **Elena Gribelyuk** [3]  **Jelani Nelson** [4,1]  **Uri Stemmer** [2,1]

## Abstract

We study algorithms in the *resettable streaming model*, where the value of each key can either be increased or reset to zero. This model is suitable for applications such as active resource monitoring with support for deletions and machine unlearning. We show that all existing sketches for this model are vulnerable to adaptive adversarial attacks that apply even when the sketch size is polynomial in the length of the stream. To overcome these vulnerabilities, we present the first adaptively robust sketches for resettable streams that require only *polylogarithmic* space complexity in the stream length. Our framework supports (sub) linear statistics including $L_p$ moments for $p \in [0, 1]$ (in particular, *Cardinality* and *Sum*) and *Bernstein statistics*. We bypass strong impossibility results known for linear and composable sketches by designing dedicated single-stream sketches robustified via Differential Privacy. Unlike standard robustification techniques, which provide limited benefits in this setting and still require polynomial space in the stream length, we leverage the *Binary Tree Mechanism* for continual observation to protect the sketch's internal randomness. This enables accurate *prefix-max* error guarantees with polylogarithmic space.

## 1. Introduction

Large-scale data systems process datasets consisting of key-value pairs $\{(x, v_x)\}$, where the number of distinct keys may be extremely large. Since it is often infeasible to store the whole dataset in memory, systems instead maintain a compact *sketch* (or *synopsis*) that supports updates to individual values and efficient real-time estimation of statistics.

In particular, the data stream consists of a sequence of operations which evolve the values of underlying keys. *Streaming sketches* are designed to process the stream of updates in a single pass, and for a chosen function $f(\cdot)$, return an accurate estimate for a statistic of the form

$$F := \sum_x f(v_x).$$

Data stream workloads originated with the development of operating systems Knuth (1998); Vitter (1985). In *unaggregated* data streams, each update *modifies* the value of a single key. The study of space-efficient sketching dates back to Morris (1978) for approximate counting, Misra and Gries (1982) for the heavy hitters problem, Flajolet and Martin (1985) for distinct counting, Gibbons and Matias (1998) for weighted sampling, and the Alon et al. (1999b) framework for estimating frequency moments.

Variants of the streaming model differ in (1) the domain of possible values of keys and (2) the way in which values evolve under updates $(x, \Delta)$. The *insertion-only* (*incremental*) model (Misra and Gries, 1982; Flajolet and Martin, 1985; Gibbons and Matias, 1998) supports non-negative updates of the form $v_x \leftarrow v_x + \Delta$ for $\Delta \geq 0$. On the other hand, the *general turnstile* model (Alon et al., 1999b) allows signed underlying values $v_x$ and signed updates $v_x \leftarrow v_x + \Delta$ for any $\Delta \in \mathbb{Z}$. Alternatively, the *strict turnstile* model allows signed updates $\Delta \in \mathbb{Z}$ under the additional *source-side* enforcement that $v_x \geq 0$ is satisfied at all times. In contrast, the ReLU model (Gemulla et al., 2006; 2007; Cohen et al., 2012) enforces the non-negativity of the values $v_x$ under signed updates by interpreting each update $(x, \Delta)$ as $v_x \leftarrow \max\{0, v_x + \Delta\}$.

In this work, we focus on the *resettable streaming model*, where values $(v_x)$ remain nonnegative and may evolve under two types of operations:

- INC$(x, \Delta)$: increment $v_x \leftarrow v_x + \Delta$ (where $\Delta \geq 0$).

- RESET$(P)$: reset $v_x \leftarrow 0$ for all keys satisfying a predicate $P$.[1]

The resettable streaming model generalizes the insertion-only model and can be viewed as a specific instance of

*Equal contribution  [1]Google Research  [2]Tel Aviv University  [3]Princeton University  [4]UC Berkeley. Correspondence to: Edith Cohen <edith@cohenwang.com>, Elena Gribelyuk <eg5539@princeton.edu>, Jelani Nelson <minilek@alum.mit.edu>, Uri Stemmer <u@uri.co.il>.

*Proceedings of the 43$^{rd}$ International Conference on Machine Learning*, Seoul, South Korea. PMLR 306, 2026. Copyright 2026 by the author(s).

---

[1]This subsumes single-key resets, denoted by RESET$(x)$, where the predicate $P$ selects only the key $x$.

the ReLU model (where $\text{RESET}(x)$ is implemented via a sufficiently large negative update $(x, -\infty)$). A fundamental property of these models is that updates are *non-commutative*, unlike in the turnstile and incremental models. This makes standard composable sketches, which are designed to support distributed processing, unnecessarily strong for our purposes. Consequently, the design goal is streaming sketches.

Semantically, reset operations augment insertion-only streaming with a natural deletion mechanism that *enforces* that only previously-inserted data can be removed, without requiring access to values. This sketch-side enforcement is crucial in settings where the original data is unavailable or impractical to access, as in streaming systems which process network traffic or large-scale training data, and in adversarial environments, where unconstrained deletion requests can be misrepresented and exploited Huang et al. (2025). The model captures the dynamics of real-time systems where values represent strictly non-negative quantities, such as active entities or resource usage. Moreover, resets naturally support compliance with modern privacy regulations by modeling explicit removal requests, including the *Right to Erasure* (Article 17 of the GDPR) (European Parliament and Council of the European Union, 2016) and the *Right to Deletion* (Cal. Civ. Code §1798.105, CCPA) (California State Legislature, 2018).

**Frequency Statistics.** We focus here on (sub)linear statistics. Two fundamental statistics are the *cardinality* ($\ell_0$ norm) of the dataset, where $f(v) := \mathbf{1}\{v \neq 0\}$, and the $\ell_1$ norm of the underlying dataset, where $f(v) = |v|$. For nonnegative values, the $\ell_1$ norm is simply the sum $\sum_x v_x$. We also consider the broad class of *soft concave sublinear* statistics (Cohen, 2016; Cohen and Geri, 2019; Pettie and Wang, 2025), where $f$ is a *Bernstein function* (Bernstein, 1929; Schilling et al., 2012). Bernstein functions include the low frequency moments ($f(v) = v^p, p \in (0, 1)$), logarithmic growth ($\ln(1 + v)$), and soft capping ($T(1 - e^{-v/T})$). Extensive prior work on sketching specific Bernstein statistics included low *Frequency Moments* ($F_p$ for $p \leq 1$) (Indyk, 2006; Kane et al., 2010), *Shannon Entropy* (Chakrabarti et al., 2006), and capping functions (Cohen, 2018). Bernstein statistics are used in applications to "dampen" the contribution of high-frequency keys in a controlled manner. Examples include frequency capping in digital advertising to maximize reach (Aggarwal et al., 2011), the weighting of co-occurrence counts in word embeddings like GloVe (Pennington et al., 2014), and the estimation of Limited Expected Value (LEV) for reinsurance pricing (Klugman et al., 2019). Dampening high contributions is also essential for privacy preservation and is used in the gradient clipping mechanism of DP-SGD (Abadi et al., 2016) and the private estimation of fractional frequency moments (Chen et al., 2022).

Cardinality is the simplest statistic in the resettable model. In particular, a resettable sketch for cardinality estimation can be obtained from a resettable sketch for any frequency-based statistic defined by a function $f$ with $f(0) = 0$ and $f(1) > 0$. This stands in contrast to the incremental model, where summation is the easiest statistic to sketch via a single counter, and to the turnstile model, where no such general reduction is known. In the resettable cardinality problem, the dataset is a dynamic set of keys, the operations are insertions and deletions of keys, and the algorithm's task is to estimate the cardinality. The reduction is given by:

$$\begin{cases} \text{INSERT}(x): & \text{RESET}(x) \text{ followed by } \text{INC}(x, 1), \\ \text{DELETE}(x): & \text{RESET}(x), \end{cases} \quad (1)$$

Since $v_x \in \{0, 1\}$ after the reduction, the statistic $\sum_x f(v_x)/f(1)$ is precisely the cardinality of the dataset. This implies that any barriers established for resettable cardinality broadly apply to any statistic.

**Prefix-max Accuracy Benchmark.** While a common accuracy guarantee for incremental or turnstile streaming algorithms is the $(1 + \varepsilon)$-relative error guarantee, it is known that any resettable streaming algorithm for cardinality estimation must use $\Omega(n)$ bits of space if required to produce a $(1 + \varepsilon)$-approximation (Pavan et al., 2024). Indeed, this follows by a standard reduction from the set-disjointness communication problem (Razborov, 1990; Kushilevitz and Nisan, 1996), by showing that any resettable cardinality sketch that outputs a $(1 + \varepsilon)$-approximation would distinguish whether two sets intersect, requiring linear space even for randomized sketches.

As a result, resettable streaming algorithms must settle for the weaker *prefix-max* error guarantee. Concretely, let $F_t$ denote the true statistic after update $t$. We seek bounds of the form

$$|\widehat{F}_t - F_t| \leq \varepsilon \max_{t' \leq t} F_{t'}, \quad (2)$$

where $\varepsilon$ decreases with sketch size. [2] Moreover, by the reduction in Eq. (1), this means that we cannot hope to design resettable streaming algorithms satisfying the relative-error guarantee for any function $f$. Fortunately, the prefix-max guarantee is sufficient for many applications: when the current value of the statistics is always at least a constant fraction of the prefix-max, and in particular, when the statistic fluctuates within a range $[f_\ell, f_h]$ with $f_h/f_\ell = O(1)$, the prefix-max error effectively translates to a relative-error approximation. Additionally (see Section 1.3), resettable with prefix-max guarantees appear fundamentally more technically challenging than relative error in incremental streams.

---

[2] Two parties have sets $A, B \subset U$. The first party applies a resettable sketch to its set $A$ and sends the sketch to the second party. The second party then resets all elements in $U \setminus B$, obtaining a sketch of $A \cap B$. A relative error approximation to the resulting cardinality would indicate whether $A \cap B$ is empty and thus solve set-disjointness on sets $A, B$.

## 1.1. The Adaptive Streaming Model

A major shortcoming of classical streaming models is that correctness analyses strongly rely on the assumption that the input stream is fixed in advance and independent of the internal randomness of the algorithm. Many practical settings require a more general model, where previous algorithm outputs may influence the future queries given to the algorithm. For instance, future database queries often depend on previous answers, and recommendation systems create feedback loops which rely on user interactions. Motivated by this gap, adaptive settings have been extensively studied in multiple areas throughout the last few decades, including statistical queries (Freedman, 1983; Ioannidis, 2005; Lukacs et al., 2009; Hardt and Ullman, 2014; Dwork et al., 2015b), sketching and streaming algorithms (Mironov et al., 2008; Hardt and Woodruff, 2013; Ben-Eliezer et al., 2021b; Hassidim et al., 2020; Woodruff and Zhou, 2021; Attias et al., 2021; Ben-Eliezer et al., 2021a; Cohen et al., 2022a;b; Ahmadian and Cohen, 2024b), dynamic graph algorithms (Shiloach and Even, 1981; Ahn et al., 2012; Gawrychowski et al., 2020; Gutenberg and Wulff-Nilsen, 2020; Wajc, 2020; Beimel et al., 2021), and adversarial robustness in machine learning (Szegedy et al., 2013; Goodfellow et al., 2014; Athalye et al., 2018; Papernot et al., 2017).

In the adaptive (or adversarial) streaming model, stream updates may depend on the entire sequence of previous updates and estimates released by the algorithm. Formally, in the resettable streaming setting, at each time $t \in [T]$, the streaming algorithm receives an adaptive update $\text{INC}(x, \Delta)$ or $\text{RESET}(x)$. We say that an algorithm is *adaptively robust* for a given function $f : \mathbb{Z} \to \mathbb{R}$ if the algorithm produces accurate estimates of $F_t = \sum_x f(v_x^{(t)})$ up to additive $\varepsilon \cdot \max_{t' \le t} F_{t'}$ error at each step $t \in [T]$, even when updates are chosen adaptively based on the transcript of previous updates and responses.

In this work, we ask the following central question:

**Question 1.1.** Can we design adaptively robust sketches for resettable streaming that output a prefix-max approximation to fundamental statistics, such as cardinality, sum, and Bernstein statistics, using $\text{poly}\left(\frac{1}{\varepsilon}, \log T, \log \frac{1}{\delta}\right)$ bits of space on a stream of length $T$?

## 1.2. Main Contribution

We answer Question 1.1 in the affirmative and design adversarially robust streaming algorithms for a large class of fundamental statistics, such as *cardinality*, *sum*, and the class of *Bernstein* (soft concave sublinear) statistics.

**Theorem 1.2** (Robust resettable cardinality, sum, and Bernstein; informal). *For any $\varepsilon, \delta \in (0, 1)$ and a resettable adaptive stream with $T_{\text{Inc}}$ increments,[3] there exist sketches*

---

[3]The bounds allow for an arbitrary number of resets.

*for cardinality, sum, and Bernstein statistics of size $k = O(\text{poly}(\varepsilon^{-1}, \log(T_{\text{Inc}}/\delta)))$ bits. With probability at least $1 - \delta$, the sketch maintains an estimate $\hat{F}_t$ satisfying the following error bound at all time steps $t$:*

$$|F_t - \hat{F}_t| \le \varepsilon \max_{t' \le t} F_{t'}.$$

We remark that our adaptively robust algorithm for $\ell_0$ is simultaneously $(\varepsilon, \delta)$ event-level differentially private in the continual observation streaming setting, where updates may be adaptively-chosen insertions or resets. This directly follows from our analysis, as each stream update affects at most two updates to the *size* of the sample maintained by our $\ell_0$ algorithm; furthermore, since our robust algorithm applies the Binary Tree Mechanism to protect updates to the sample-size, this also protects individual updates as a byproduct.

## 1.3. Review of Prior Work

We start with a concise review of the literature and gained insights. For additional details, see Section A.

The resettable streaming model (as well as the more general ReLU model) were introduced by (Gemulla et al., 2006; 2007) for $\ell_1$ estimation for unit updates and was later extended by Cohen et al. (2012) for weighted updates. The resettable model was (re)-introduced in Pavan et al. (2024). Lin et al. (2025) presented algorithms for frequency moments estimation $f(v) = |V|^p$ for $p \in (0, 2]$.[4] These works design streaming algorithms which satisfy the prefix-max error guarantee and use $\text{poly}\left(\frac{1}{\varepsilon}, \log T, \log \frac{1}{\delta}\right)$ bits of memory. The core component of each algorithm is a *sampling sketch*, which maintains a weighted sample of the keys. Sampling sketches are naturally suitable for the resettable streaming model, as a key reset can be implemented by simply deleting that specific key from the sample if it was included, and if it was not included then no action is needed. However, we observe that all prior sampling-based algorithms are vulnerable to adaptive inputs: Intuitively, if the output of the sampling-based algorithm changes only when a key is added to the sample, the adversary can then strategically reset that key, thus creating a bias in the estimation. Crucially, this attack exploits the fact that sample membership is revealed through changes in the current estimate.

The core difficulty in designing adaptively robust streaming algorithms is that the algorithm's internal randomness needs to be periodically "refreshed" or concealed from the adaptive adversary. *Wrapper methods* are black-box approaches

---

[4]The model was introduced with different names. We chose to use *resettable* streaming to emphasize that the model captures a wider class of applications than unlearning, find the term better than *rewrites* (Cohen et al., 2012), and less confusing than *deletions* (Gemulla et al., 2006; 2007; Cohen et al., 2012) (that are also used to refer to turnstile negative updates).

that take as input a generation process for a sketch in the non-adaptive setting and output a robust sketch which has correctness guarantees in the adaptive setting. For instance, the wrapper methods of (Hassidim et al., 2020; Blanc, 2023) instantiate and maintain $k$ independent non-adaptive sketches and leverage techniques from differential privacy to privately release aggregated estimates which conceal each sketch's internal randomness, allowing for $T = O(k^2)$ adaptive interactions. The *sketch switching* wrapper Ben-Eliezer et al. (2021b) maintains $k$ independent sketches and uses a new sketch each time that the estimate changes significantly (and thus exposes its randomness). However, in contrast to the incremental and *bounded deletions* model, where the number of needed changes is logarithmic in $T$, in the resettable streaming model, the statistic can oscillate between $\min_t F_t$ and $\max_t F_t$ $\tilde{\Theta}(T)$ times. Therefore, previous approaches do not facilitate our goal of designing an algorithm which uses polylogarithmic space in $T$.

On the other hand, adaptive attacks on cardinality sketches in the turnstile model and query-model by Ahmadian and Cohen (2024a); Cohen et al. (2024) transfer to resettable streaming (with prefix-max error guarantees). Combining this observation with Eq. (1), any robust resettable streaming sketch that is union-composable or linear requires a sketch size (or sketch dimension) that is polynomial in $T$. To circumvent this lower bound, our robust polylogarithmic-size sketches are not composable.

### 1.4. Overview and Road Map

Our robust sketches for cardinality and sum are obtained by a careful redesign and robustification of the *non-robust* resettable sampling sketches of (Gemulla et al., 2006; 2007; Cohen et al., 2012; Pavan et al., 2024), which in turn are based on incremental streaming sampling sketches Gibbons and Matias (1998); Estan and Varghese (2002). We leverage techniques from differentially privacy including its generalization property (Dwork et al., 2015a; Bassily et al., 2021; Feldman and Steinke, 2018) and enhanced applications of *continual reporting* using the *tree mechanism* (Chan et al., 2011; Dwork et al., 2010; Dwork and Roth, 2014). The tree mechanism, in particular, has not previously been applied to robustify sampling sketches. In contrast to standard DP-based robustification methods such as Hassidim et al. (2020) (based on (Dwork et al., 2015b; Bassily et al., 2021)), which achieve only a quadratic improvement in robustness, we obtain an *exponential* improvement.

Our results tightly dodge the barriers observed in Section 1.3: We circumvent lower bounds on composable and linear sketching by building on dedicated streaming sampling sketches. Our sketches are for (sub) linear statistics, noting that there are no known sampling sketches for super-linear statistics without persistent randomness (which is

known to create adaptive vulnerability).

We instructively start with the cardinality estimation problem, as it provides a simple setting to introduce our robustification techniques. In Section 2 we review the standard cardinality with deletions sketch (Gemulla et al., 2006; 2007; Pavan et al., 2024).[5] where the sampling rate $p$ is fixed throughout the execution. In non-adaptive streams, the sketch maintains an i.i.d. Bernoulli sample of the distinct keys. We show that even this basic variant is not robust to adaptive inputs. We then introduce a robust estimator to the fixed-rate sketch by composing the algorithm with the Binary Tree Mechanism on the stream of reported estimates (See analysis in Section B). Finally, we provide an overview of our adjustable-rate robust sketch and estimator that adaptively decreases the sampling rate over time to ensure that the sample size (and therefore the sketch size) remains bounded and attains the prefix-max Eq. (2) error guarantee (See analysis in Section C).

In Section 3 (analysis in Section D) we present our robust resettable sum sketch. We review the standard fixed-rate resettable sum sketch by Cohen et al. (2012) (extends the unit-updates sketch by Gemulla et al. (2006)) and show it is not robust with the standard estimator. We present a robust estimator for the fixed-rate sketch and a robust adjustable rate sketch and estimator. While our algorithms are simple to implement, the analysis required overcoming some technical hurdles including identifying appropriate randomness units to control sensitivity and facilitate generalization and a modified analysis of the tree mechanism.

In Section 4 we preview our design of robust resettable sketches for Bernstein (soft concave sublinear) statistics (full details and analysis are in Section E). We build on the composable sketching framework of Cohen (2016) and "reduce" the resettable sketching of any Bernstein statistics to resettable sketching of sum and cardinality statistics.

## 2. Robust Resettable $\ell_0$ estimation

The *Cardinality with Deletions Streaming Problem* tracks an evolving set of active keys $A_t \subseteq U$ under a stream of updates $E = (\mathsf{op}_1, \ldots, \mathsf{op}_T)$, where each update is either $\mathsf{Insert}(x)$, which adds $x$ to $A_t$, or $\mathsf{Delete}(x)$, which removes $x$ from $A_t$. Our goal is to maintain a compact sketch and publish an estimate $\hat{N}_t$ of $N_t = |A_t|$. In non-adaptive settings, we seek guarantees that hold when the sequence of updates does not depend on the published estimates. In adaptive settings, we seek guarantees that hold when each update $\mathsf{op}_t$ may depend on the history of reported estimates

---

[5]The sketch is equivalent to transforming the resettable cardinality stream to a resettable sum stream via Eq. (1) and applying the ReLU sum sketch of Gemulla et al. (2006; 2007). An insertions-only version was presented in (Chakraborty et al., 2022).

**Algorithm 1:** Bernoulli Cardinality with Deletions (fixed $p$) (Gemulla et al., 2006; 2007; Pavan et al., 2024)

**Input:** Stream of $\mathsf{Insert}(x)$ or $\mathsf{Delete}(x)$; sampling prob $p$

**State:** Set $S$ of sampled active keys (initially $\emptyset$).

**for** $t = 1, 2, \ldots$ **do**
    Receive operation on key $x$
    **if** $\mathsf{Insert}(x)$ **then**
        $S \leftarrow S \setminus \{x\}$    // refresh sample status
        Sample $b \sim \mathsf{Bernoulli}(p)$
        **if** $b = 1$ **then** $S \leftarrow S \cup \{x\}$
    **else if** $\mathsf{Delete}(x)$ **then**
        $S \leftarrow S \setminus \{x\}$    // remove if present
    $\hat{N}_t \leftarrow |S|/p$; **return** $\hat{N}_t$    // cardinality estimate
      of active keys

$\{\hat{N}_1, \ldots, \hat{N}_{t-1}\}$. Cardinality with deletions is a resettable streaming problem with the $\ell_0$ norm $f(v) := \mathbf{1}\{v > 0\}$, through the equivalence $\mathsf{Insert}(x) \equiv \mathsf{Inc}(x, \Delta)$ $(\Delta > 0)$ and $\mathsf{Delete}(x) \equiv \mathsf{Reset}(x)$.

In Section 2.1 review the standard sketch (Gemulla et al., 2006; Pavan et al., 2024) and in Section 2.2 we show that the standard sketch with the standard estimator is vulnerable to adaptive updates. In Section 2.3 we overview our robust sketch designs, with analysis deferred to the appendix.

### 2.1. Standard Cardinality with Deletions Sketch

We review the standard Bernoulli sampling sketch (Gemulla et al., 2006; Chakraborty et al., 2022; Pavan et al., 2024), presented in Algorithm 1.

In the non-adaptive setting, $S_t$ is a Bernoulli sample where each active key in $A_t$ is included with probability $p$. Therefore,

$$\hat{N}_t = \frac{|S_t|}{p} \tag{3}$$

is an unbiased estimator of $N_t$.

When a key $x$ is inserted at time $t$, the algorithm first removes $x$ from $S_t$ if it is present and resamples $x$ into the sample independently with probability $p$. This preserves the property that the active key $x$ is included in $S_t$ independently with probability exactly $p$. When a key $x$ is deleted, the algorithm simply removes $x$ from the sample if it is present, and does nothing otherwise.

**Lemma 2.1** (Guarantees for non-adaptive streams)**.** *For any desired accuracy $\varepsilon \in (0, 1)$ and confidence $\delta \in (0, 1)$, if we apply Algorithm 1 with*

$$p = \frac{3}{\varepsilon^2 N_{\max}} \log\left(\frac{2T}{\delta}\right),$$

*on a non-adaptive stream with $\max_{t \leq T} N_t \leq N_{\max}$, then with probability at least $1 - \delta$,*

$$\forall t \in [T], \qquad |\hat{N}_t - N_t| \leq \varepsilon N_{\max}$$
$$|S_t| \leq 3(1 + \varepsilon)\varepsilon^{-2} \log(2T/\delta).$$

**Remark 2.2** (Prefix-max error guarantee via rate adjustments for the standard sketch)**.** A standard refinement of the Bernoulli sketch adaptively decreases the sampling rate so as to keep the sample size bounded (Gemulla et al., 2007; Cohen et al., 2007; Chakraborty et al., 2022). Whenever $|S_t|$ exceeds a fixed threshold $k$, the algorithm halves the sampling rate $p \leftarrow p/2$ and independently retains each key in $S_t$ with probability $1/2$. This preserves the invariant that every distinct active key is present in the sample with probability $p$. This scheme yields a prefix-max error bound: for a choice of $k = O(\varepsilon^{-2} \log(T/\delta))$, one obtains

$$\forall t \in [T], \qquad |\hat{N}_t - N_t| \leq \varepsilon \cdot \max_{t' \leq t} N_{t'}.$$

### 2.2. Vulnerability to Adaptive Attacks

While Algorithm 1 is unbiased for non-adaptive streams, we observe that it is catastrophically vulnerable to adaptive adversaries.

**Claim 2.3** (Failure of standard estimator for Algorithm 1)**.** There exist adaptive adversaries such that:

1. **(Insertion-Only)** In an insertion-only stream with $N$ distinct keys, the estimator $\hat{N}_T$ underestimates the true count by a factor of $p$ (i.e., $\mathbb{E}[\hat{N}_T] = p \cdot N$).

2. **(Resettable)** In a resettable stream with $T$ updates where the true cardinality is $N_T = \Theta(T)$, the estimator reports $\hat{N}_T = 0$ with probability 1.

#### 2.2.1. THE RE-INSERTION ATTACK (INSERTION-ONLY)

Even without deletions, an adversary can bias the sample probability from $p$ to $p^2$ simply by re-inserting keys that are successfully sampled: For each new key $x$, the adversary issues $\textsc{Insert}(x)$. If the estimate $\hat{N}_t$ increases (implying $x \in S$), the adversary immediately issues $\textsc{Insert}(x)$ again. It then never touches $x$ again.

**Analysis.** Recall that Algorithm 1 handles a re-insertion by removing the existing copy and resampling. If $x$ is not sampled initially (prob $1 - p$), it remains out. If $x$ is sampled initially (prob $p$), it is re-inserted and effectively re-sampled with probability $p$. The final inclusion probability is $\Pr[x \in S] = p \cdot p = p^2$. Since the estimator scales the sample size by $1/p$, the expected estimate is $(N \cdot p^2)/p = pN$. For small $p$ (e.g., $1\%$), the error is massive.

## 2.2.2. SAMPLE-AND-DELETE ATTACK (RESETTABLE)

With access to deletions, the adversary can completely empty the sample while keeping the active set large, with the following strategy. For each step $t$:

1. The adversary inserts a new key $x_t$.

2. If $\hat{N}_t > \hat{N}_{t-1}$ (implying $x_t \in S_t$), the adversary immediately issues DELETE$(x_t)$.

3. If $\hat{N}_t$ does not change (implying $x_t \notin S_t$), the adversary leaves $x_t$ active.

**Analysis.** Any key that enters the sample is immediately deleted. Any key that fails the sampling coin flip remains active. Consequently, at time $T$, the sample is empty ($S_T = \emptyset \implies \hat{N}_T = 0$), while the true set $A_T$ contains all keys that were rejected by the filter. The true cardinality is $N_T \approx (1-p)T$, resulting in infinite relative error. Such bias could also arise in non-malicious settings due to feedback. For example, a load balancing system might attempt to reassign load when the estimated load increases.

**Intuition.** The vulnerability arises because observing the estimate $\hat{N}_t$ gives the adversary information *better than the prior $p$* about whether a given active key is present in the sample. We robustify by ensuring a posterior that is close to the prior.

### 2.3. Robustification

**Theorem 2.4** (Robust Cardinality with Deletions). *For any accuracy parameter $\varepsilon \in (0,1)$ and confidence $\delta \in (0,1)$, there exists a streaming sketch for cardinality with deletions which, under an adaptive update sequence of length $T$, maintains estimates $(\hat{N}_t)_{t \leq T}$ using space[6]*

$$O\left(\frac{1}{\varepsilon^2} \log^{3/2} T \cdot \log \frac{T}{\delta}\right).$$

*With probability at least $1 - \delta$, the estimator satisfies the* uniform *prefix-max error guarantee*

$$\forall t \in [T], \qquad |\hat{N}_t - N_t| \leq \varepsilon \max_{t' \leq t} N_{t'}. \tag{4}$$

*In particular, since $\forall t \in [T], \max_{t' \leq t} N_{t'} \leq N_{\max}$, where $N_{\max} = \max_{t \leq T} N_t$, the estimator satisfies the uniform overall-max error guarantee:*

$$\forall t \in [T], \qquad |\hat{N}_t - N_t| \leq \varepsilon N_{\max}. \tag{5}$$

---

[6]The space bound applies with $T_{\mathrm{Inc}}$, which is the number of INC operations. This by observing that in the robustness analysis we can ignore resets performed on keys with $v_x = 0$ and that the total number of applicable reset operations is bounded by $T_{\mathrm{Inc}}$ (hence effectively for our analysis $T \leq 2T_{\mathrm{Inc}}$).

---

**Algorithm 2:** Robust Adjustable-Rate Cardinality

**Input:** Stream of Insert$(x)$ or Delete$(x)$ operations; initial sampling probability $p_0 \leftarrow 1$; sample-size budget $k$; error margin $\alpha$; confidence parameter $\delta$; DP tree mechanism Tree (Section G) with parameter $(T_{\mathrm{tree}} = T + O(\log T), L = 2, \varepsilon_{\mathrm{dp}})$

**State:** Sample $S$ of active keys (initially $\emptyset$)
Sampling probability $p \leftarrow p_0$
Current sample size $s \leftarrow 0$; previous size $s_{\mathrm{prev}} \leftarrow 0$
Internal state of Tree initialized to all zeros

**for** $t = 1, 2, \ldots, T$ **do**
  receive operation on key $x$
  **if** Insert$(x)$ **then** // Update sample under current p
    **if** $x \in S$ **then** $S \leftarrow S \setminus \{x\}; s \leftarrow s - 1$
    draw $b \sim$ Bernoulli$(p)$
    **if** $b = 1$ **then** $S \leftarrow S \cup \{x\}; s \leftarrow s + 1$
  **else if** Delete$(x)$ **then**
    **if** $x \in S$ **then** $S \leftarrow S \setminus \{x\}; s \leftarrow s - 1$
  // Feed sample-size updates to tree mechanism
  $u_t \leftarrow s - s_{\mathrm{prev}}; s_{\mathrm{prev}} \leftarrow s$
  $\tilde{S}_t \leftarrow$ Tree.UPDATEANDREPORT$(u_t)$
  // Possibly adjust sampling rate
  **while** $\tilde{S}_t > k - \alpha$ **do** // halve rate and subsample
    $p \leftarrow p/2; s_{\mathrm{old}} \leftarrow s; S' \leftarrow \emptyset$
    **foreach** $x \in S$ **do**
      draw $b' \sim$ Bernoulli$(1/2)$
      **if** $b' = 1$ **then** $S' \leftarrow S' \cup \{x\}$
    $S \leftarrow S'; s \leftarrow |S|$
    // report sample-size change to tree mech.
    $u^{\mathrm{adjust}} \leftarrow s - s_{\mathrm{old}}$
    $s_{\mathrm{prev}} \leftarrow s$
    $\tilde{S}_t \leftarrow$ Tree.UPDATEANDREPORT$(u^{\mathrm{adjust}})$
  **if** $p = 1$ **then return** $|S|$ **else return** $\hat{N}_t \leftarrow \tilde{S}_t/p$

**Fixed-rate Robust Design.** For instructive reasons, we first present a construction and analysis of a basic fixed-rate robust sketch that assumes prior bound on $N_{\max}$ and gives the overall-max error guarantees Eq. (5). This construction composes the standard fixed-rate sketch and estimator from Algorithm 1 with the Binary Tree Mechanism for continual release (Chan et al., 2011; Dwork et al., 2010; Dwork and Roth, 2014) (as described and analyzed in Section G with sensitivity parameter $L = 2$ and privacy parameter $\varepsilon_{\mathrm{dp}}$).

The mechanism input is a stream of updates to the *sample size* and its output is a sequence of noisy approximate prefix sums, which are estimates $(\tilde{S}_t)$ of the sample size. Our algorithm returns estimates given by $\hat{N}_t := \frac{\tilde{S}_t}{p}$. Intuitively, this DP layer facilitates robustness by protecting the information on the presence of each key in the sample so that the posterior knowledge of the adversary remains close to the prior $p$. The analysis of the construction is in Section B.

**Adjustable-rate Robust Sketch Design.** Our adjustable-rate algorithm is described in Algorithm 2, with parameter settings and full analysis given in Section C. We establish

that it achieves prefix-max guarantees Eq. (4) and does not require a prior bound on $N_{\max}$. In contrast to the non-robust adjustable design Remark 2.2, which uses exact sample size to trigger rate adjustments $p \to p/2$, the robust sketch uses interactively the noisy output of the tree mechanism: When the estimate returned approaches the sample size bound $k$ we update $p \leftarrow p/2$, subsample $S_t$ accordingly, and issue respective sample-size update to the tree mechanism.

## 3. Robust Resettable $\ell_1$ Estimation

Next, we consider the problem of estimating the sum $F_t = \sum_x v_x^{(t)}$ under adaptive increment and reset operations.

**Theorem 3.1** (Robust Resettable Sum Sketch). *Given accuracy parameter $\varepsilon > 0$ and failure probability $\delta > 0$, there exists an adversarially robust sketch for the sum $F = \sum_x v_x$ under a stream of $T$ increments and reset operations that, with probability at least $1 - \delta$, outputs an estimate $\hat{F}_t$ such that*

$$|\hat{F}_t - F_t| \le \varepsilon \cdot \max_{t' \le t} F_t, \quad \text{for all } t \in [T] \qquad (6)$$

*The algorithm uses $O\left(\frac{1}{\varepsilon^2} \log^{11/2}(T) \cdot \log^2\left(\frac{1}{\delta}\right)\right)$ bits of memory.*

We first present the non-robust algorithm of Cohen et al. (2012) (simplified unit updates by Gemulla et al. (2006)).

---

**Algorithm 3:** Resettable Sum Sketch (fixed $\tau$) (Cohen et al., 2012)

---

**Input:** Stream of updates on keys $i$ of two types: $\mathsf{Inc}(i, \Delta)$ with $\Delta \ge 0$ and $\mathsf{Reset}(i)$; sampling threshold $\tau$
**State:** Sample $S$ of keys; each key $i \in S$ has counter $c_i$.
**Output:** At each time $t$, an estimate of the resettable streaming sum

**foreach** *update on key $i$* **do**
  **if** $\mathsf{Inc}(i, \Delta)$ *with $\Delta > 0$* **then**
    **if** $i \in S$ **then**
      $c_i \leftarrow c_i + \Delta$
    **else** // case $i \notin S$
      sample $r \sim \mathrm{Exp}[\tau^{-1}]$    // mean $\tau$, rate $1/\tau$
      **if** $r < \Delta$ **then** $S \leftarrow S \cup \{i\}; c_i \leftarrow \Delta - r$
      // start tracking $i$
  **else if** $\mathsf{Reset}(i)$ **then**
    **if** $i \in S$ **then** $S \leftarrow S \setminus \{i\}$
    // if $i \notin S$, nothing to do
  **return** $\sum_{i \in S}(\tau + c_i)$      // estimate of the sum

---

**Vulnerability.** Algorithm 3 can implement the cardinality with deletions in Algorithm 1 (set $\tau = -1/\log(1 - p)$, implement $\mathsf{Insert}(x)$ by $\mathsf{Reset}(x)$; $\mathsf{Inc}(x, a)$, implement $\mathsf{Delete}(x)$ by $\mathsf{Reset}(x)$). Therefore the attack described

in Section 2.2 shows that the algorithm is not robust to adaptive updates.

**Alternative View of Algorithm 3.** We will work with an equivalent view of Algorithm 3 that casts it as a deterministic function of the input stream and a set of fixed, independent random variables (entry thresholds), with independent contributions to the estimate and sample.

**Definition 3.2** (Entry-Threshold Formulation). For each key $x$ and each reset epoch $j$, let $R_{x,j} \sim \mathrm{Exp}(1/\tau)$ be an independent *entry threshold*. Let $v_x^{(t)}$ denote the active value of key $x$ at time $t$ (accumulated since the last reset). The state of the sketch at time $t$ is determined strictly by the pairs $\{(v_x^{(t)}, R_{x,j})\}$:

1. **Sample Indicators:** The presence of key $x$ in the sample is given by the indicator variable $I_x^{(t)}$. Key $x$ is sampled if and only if its value exceeds its active threshold:

$$I_x^{(t)} := \mathbb{I}\left\{v_x^{(t)} > R_{x,j}\right\}.$$

   The total sample size is the sum of these independent indicators:

$$|S_t| = \sum_x I_x^{(t)}.$$

2. **Sum Estimator:** The contribution of key $x$ to the sum estimate is defined as:

$$Z_x^{(t)} := I_x^{(t)} \cdot \left(v_x^{(t)} + \tau - R_{x,j}\right).$$

   The total estimate is the sum of these independent contributions:

$$\hat{F}_t = \sum_x Z_x^{(t)}.$$

Note that other important properties of Algorithm 3 are analyzed in Section D.1. Since the entry threshold $R_{x,j}$ determines both the *sample membership* and the *estimated contribution* of $x$ to $\hat{F}_t$, intuitively we must ensure that final algorithm outputs effectively hide information about all *currently active* entry thresholds $\{R_{x,j}\}$. However, this cannot be achieved by composing the algorithm with the Binary Tree Mechanism directly (as we did for cardinality estimation), as the sensitivity of updates to $\hat{F}_t$ may be unbounded. To this end, we observe that with probability $1 - \delta$, the following event holds for $B := \tau \log(T/\delta)$:

$$\mathcal{E} = \{c_x^{(t)} \ge v_x^{(t)} - B \text{ for all keys } x, \, t \in [T]\}$$

In particular, for all $t \in [T]$, if a key $x$ has a large value $v_x^{(t)} > B$, then $x$ is deterministically in the sample, with

contribution $c_x^{(t)} \geq v_x^{(t)} - B$. Then, intuitively, after conditioning on $\mathcal{E}$, we only need to protect the randomness of keys $x$ whose membership in the sample is still uncertain.

Concretely, we can separate the contribution of each key $x$ to $F_t$ into a *protected* component, which has bounded sensitivity, and a *deterministic* component, which is known to the adversary. Namely, we decompose $F_t$ as follows:

$$F_t = \sum_x \max(v_x^{(t)} - B, 0) + \sum_x \min(B, v_x^{(t)}) = D_t + P_t$$

and similarly the estimator $\hat{F}_t$ is decomposed as

$$\hat{F}_t = \sum_x \max(v_x^{(t)} - R_{x,j} + \tau - B, 0)$$
$$+ \sum_x \min(v_x^{(t)} - R_{x,j} + \tau, B) = \hat{D}_t + \hat{P}_t.$$

**Fixed-rate Robust Algorithm.** Unlike the cardinality algorithm, a key technical challenge is that each private unit $(x, j, R_{x,j})$ will contribute to many distinct updates to $\hat{P}_t$, from the moment of the last $j$-th reset to the time when $v_x^{(t)}$ exceeds $B$. Our robust algorithm then feeds updates $u_t = \frac{1}{B}\left(\hat{P}_t - \hat{P}_{t-1}\right)$ to the Binary Tree Mechanism, which returns a sequence of noisy prefix-sums $\tilde{U}_t$. Note that the Binary Tree Mechanism is re-analyzed for the setting where each unit has a bounded *total contribution* but may contribute in many time steps (for us, the total contribution of $(x, j, R_{x,j})$ to $\hat{P}_t$ is at most 2). Then, the output of our algorithm is given by $\tilde{F}_t = B \cdot \tilde{U}_t + \hat{D}_t$. The complete analysis of the fixed-rate sketch is in Section D.3 and Section D.4. Finally, to achieve error which depends on the prefix-max, we simply run $O(\log T)$ instances of the robust algorithm at different input scales; see Section D.5 for the complete algorithm description and analysis.

## 4. Robust Resettable Bernstein Sketch

We construct robust resettable sketches for any $f$ that is a *Bernstein function*, a class of sublinear functions rooted in the work of Bernstein (1929) and Schilling et al. (2012).

**Definition 4.1** (Bernstein Functions). We denote by $\mathcal{B}$ (also referred to as the *soft concave sublinear* in (Cohen, 2016; Cohen and Geri, 2019)) the set of all functions $f(w)$ that admit a Lévy-Khintchine representation with zero drift. Formally, $f \in \mathcal{B}$ if there exists a non-negative weight function $a(t) \geq 0$, referred to as the *inverse complement Laplace transform*, such that:

$$f(w) = \mathcal{L}^c[a](w) := \int_0^\infty a(t)\left(1 - e^{-wt}\right) dt.$$

We establish the following:

**Theorem 4.2** (Robust Resettable Sketches for Bernstein Statistics). *Let $f \in \mathcal{B}$ be a Bernstein function. For any accuracy parameter $\varepsilon \in (0, 1)$, failure probability $\delta \in (0, 1)$, a parameter $T$ that bounds the number of* Inc *operations, and a fixed polynomial so that update values are in a range $\Delta \in [\Delta_{\min}, \Delta_{\max}]$ where $\Delta_{\max}/\Delta_{\min} = O(\text{poly}(T))$. There exists a resettable streaming sketch using space*

$$k = O\left(\text{poly}(\varepsilon^{-1}, \log(T/\delta))\right)$$

*that, with probability at least $1 - \delta$ against an adaptive adversary, maintains an estimate $\hat{F}_t$ satisfying for all time steps $t$:*

$$|F_t - \hat{F}_t| \leq \varepsilon \max_{t' \leq t} F_{t'}.$$

The sketch construction and analysis are provided in Section E. At a high level, we show that Bernstein statistics on a resettable stream can be approximated to within a relative error by a linear combination of sum and cardinality statistics of the following form:

$$\tilde{F}(W) = \alpha_0 \cdot \mathsf{Sum}(W) + \frac{1}{r} \sum_{i \in [m]} \alpha_i \mathsf{Distinct}(E_i).$$

where the coefficients $\alpha_i$ depend on $T, \varepsilon, \delta, f(\cdot)$, $m$ is logarithmically bounded, $\mathsf{Sum}$ is applied to the input stream and the distinct statistics are computed for new resettable cardinality streams $(E_i)_{i \in [m]}$ that are generated element by element by a randomized map (as in Algorithm 5) from the input stream.

We build on the composable sketching framework of Cohen (2016) that essentially reduces sketching Bernstein statistics to sketching sum and max-distinct statistics (generalized cardinality). To facilitate robustness, we introduced a new parametrization for the composition so that the coefficients do not depend on the sum statistic value and expressed the composition using plain cardinality statistics instead of max-distinct statistic (for which there are no direct known resettable sketches). Crucially, each cardinality output stream is generated with independent randomness, so we were able to "push" the randomness introduced in the reduction map into the robustness analysis of the cardinality sketches. Finally, the original work only facilitated incremental datasets and we needed to carefully integrate support for reset operations so that robustness and prefix-max guarantees transfer.

## 5. Conclusion

We have presented the first adaptively robust streaming algorithms for the resettable model, covering (sub)linear statistics, as well as the first resettable sketches for non-moment Bernstein statistics (e.g., soft capping) that achieve polylogarithmic space complexity.

We conclude with several interesting directions for future work. First, the existence of robust resettable sketches for super-linear statistics (e.g., $\ell_2$) remains open. The core obstacle is that current sampling sketches all rely on persistent per-key randomness which inherently makes sketches vulnerable to adaptive attacks. Progress on this direction would require a streaming-specific fundamental departure from known sampling designs. Another open question is to design robust sketches in the general ReLU model (e.g. for $\ell_1$ estimation). In this setting, the central challenge is bias amplification: while resets provide a way to "clear" adversarial bias, partial decrements in ReLU may cause slight posterior biases to persist and accumulate as keys oscillate in and out of the sample. Finally, we optimized for clarity, and the exact space bounds can be tightened: for example, analysis of the tree mechanism via approximate DP or zCDP would directly improve space complexity, and the Bernstein reduction likely admits further optimization.

## Impact Statement

This paper presents work whose goal is to advance the field of Machine Learning. There are many potential societal consequences of our work, none which we feel must be specifically highlighted here.

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

# A. Detailed Prior Work and Landscape Review

## A.1. Prior Work in the Resettable and ReLU Streaming Models

We review previous non-robust algorithms for cardinality and $\ell_1$ estimation in the ReLU and resettable streaming models. While all known streaming algorithms in the turnstile model are *linear sketches*[7], unfortunately linear sketches are not designed to handle non-linear ReLU and reset operations. Consequently, all existing resettable sketches are *sampling sketches*, which maintain a weighted sample of active keys as well as their corresponding estimated (or exact) values, in addition to any sampling parameters needed for unbiased estimation. Sampling-based sketches are a natural choice when designing streaming algorithms for the resettable or ReLU models: if a $\text{RESET}(x)$ operation arrives in the stream and $x$ is not sampled by the algorithm, then no action is needed; otherwise, if $x$ is in the sample, it is removed. More generally, batch reset operations can be processed by simply deleting all keys in the sample that satisfy a given predicate.

The ReLU streaming model (as streaming with deletions model) was introduced by Gemulla et al. (2006; 2007), who focused on $f(v) = v$ with unit updates and adapted the sampling sketches of Gibbons and Matias (1998); Estan and Varghese (2002), which maintain weighted samples of active keys and obtained unbiased estimators. This algorithm was later extended by (Cohen et al., 2012) to support weighted updates and seamless adjustment of the effective sampling rate, by specifying an unaggregated implementation of the *ppswor* (probability proportional to size without replacement) sampler (Rosén, 1997; Cohen, 1997; Cohen and Kaplan, 2007; Efraimidis and Spirakis, 2006) and adding deletion support. More recently, resettable streaming has been studied with the motivation of unlearning and the right to erasure/deletion regulations (European Parliament and Council of the European Union, 2016; California State Legislature, 2018), as support to user-deletion requests in machine learning and data systems (Pavan et al., 2024). This line of work was recently extended by (Lin et al., 2025), which presented resettable sketches for $\ell_p$ statistics $f(v) = v^p$ for $p \in (0, 2]$.

## A.2. Landscape Exploration

The core difficulty in designing adaptively robust streaming algorithms is that the algorithm's internal randomness needs to be periodically "refreshed" or concealed from the adaptive adversary. Otherwise, if the algorithm samples all of its random bits at the beginning of the execution and is deterministic thereafter, the adaptive adversary may learn information about the internal randomness of the sketch and then choose adaptive queries which are highly correlated with the internal randomness, leading the sketch to fail to provide accurate estimates with high probability.[8] In fact, this is exactly the approach taken in previous lower bounds for various classes of sketching algorithms (Ahmadian and Cohen, 2024a; Cohen et al., 2024; Gribelyuk et al., 2024; 2025), and a similar "information leak" phenomenon lies behind many known lower bounds in adaptive data analysis.

Adaptively robust algorithms can be designed by carefully hiding the internal randomness of the algorithm from the adversary. Over the last decade, this was done by designing (black-box) wrapper methods which take as input a generation process for a sketch in the non-adaptive setting, and output a robust sketch which has correctness guarantees in the adaptive setting (Dwork et al., 2015b; Bassily et al., 2021; Hassidim et al., 2020; Blanc, 2023). For example, the robustness wrapper of (Hassidim et al., 2020) instantiates $k$ independent non-adaptive sketches which are maintained throughout the stream, and for each query, combines the estimates from each sketch to produce a single *privatized* estimate which effectively protects the random bits of each of the $k$ independent sketches. A different approach by (Blanc, 2023) achieves the same robustification by only using subsampling. Although sketches in non-adaptive models can typically handle exponentially many queries in the sketch size, it has been shown that these robustification techniques can be used to design algorithms that are accurate for only $O(k^2)$ adaptive interactions. Thus, this flavor of wrapper methods cannot be used to provide a positive answer to Question 1.1.

In the special case of insertion-only streaming, the space complexity of streaming algorithms under adaptive updates is relatively well-understood, as it is known that many fundamental streaming problems admit robust algorithms using sublinear space (Woodruff and Zhou, 2021; Braverman et al., 2021). Unfortunately, the techniques used to design adaptively robust algorithms in the insertion-only model do not appear to transfer to the resettable model. For instance, MinHash cardinality

---

[7]Let $x \in \mathbb{Z}^n$ denote the frequency vector of updates to the dataset. A linear sketch maintains $Ax \in \mathbb{R}^r$ for $r = o(n)$, where $A$ is a carefully designed matrix which can be efficiently stored. At the end of the stream, the algorithm returns $f(Ax)$ for a chosen post-processing function $f$.

[8]Although adaptive inputs may be benign (as in feedback-driven system behavior) or adversarial (intentional malicious manipulation), in either case, the update sequence may become correlated with the internal randomness of streaming algorithm and thus cause the randomized algorithm to fail with high probability.

sketches (Flajolet and Martin, 1985; Cohen, 1997; Flajolet et al., 2007; Heule et al., 2013) with the standard estimator (and full or cryptographically protected randomness) are fully robust in insertion-only streams but cannot support reset operations. Additionally, a common way to "robustify" streaming algorithms is to use the *sketch switching* technique, which was first introduced by Ben-Eliezer et al. (2021b). Suppose the algorithm's task is to estimate the value of a monotone, non-decreasing function $f$ throughout the stream. If the stream is insertion-only, the value of $f$ will increase by a factor of $(1 + \varepsilon)$ at most $B = O(\varepsilon^{-1} \log \max_t F_t / \min_t F_t)$ times. Then, to design an adversarially robust streaming algorithm for $f$, we can simply maintain $B$ copies of the non-robust sketch, and use a single copy at any time: once the (true) function value changes by $(1 + \varepsilon)$ compared to the last estimate returned by the current active instance, we can switch to the next instance, thereby protecting the internal randomness of each instance. Importantly, observe that this general approach can be used to robustify a streaming algorithm for $f$ only if we can upper bound the number of significant changes to the function value. As such, sketch switching can be used to robustify algorithms in the "bounded deletions" model (Ben-Eliezer et al., 2021b), which supports deletions but guarantees error proportional to the statistic on the absolute values stream, yielding a much weaker guarantee than prefix-max error that allows for a logarithmic bound on the number of switches. However, in the resettable streaming model, also with the prefix-max error guarantees, the statistic can oscillate between $\min_t F_t$ and $\max_t F_t$ $B = \tilde{\Omega}(T)$ times. Therefore, sketch switching cannot be used to provide a positive answer to Question 1.1.

On the other hand, negative results have been established for specific families of sketches by designing adaptive sequences of queries which allow the adversary to gradually learn the random bits used by the algorithm [9]. The efficacy of the attack is often measured by the *number of adaptive queries* required to break a sketch of a given *size $k$*. Over the last few decades, there has been a flurry of works which presented attacks using $\text{poly}(k)$ queries on sketches of size $k$ for $L_p$ estimation; in fact, for linear sketches for $L_2$ estimation and composable sketches for cardinality estimation, the adaptive attacks even achieved optimal $O(k^2)$ query complexity (Hardt and Woodruff, 2013; Hardt and Ullman, 2014; Steinke and Ullman, 2015; Cohen et al., 2022b; Ahmadian and Cohen, 2024b; Gribelyuk et al., 2024; Cohen et al., 2024; Gribelyuk et al., 2025), with Fingerprinting Codes (Boneh and Shaw, 1998) at the core of all of these attacks. Moreover, several well-known sketches have been shown to admit *linear* size adaptive attacks. In particular, Cherapanamjeri and Nelson (2020) showed that the Johnson Lindenstrauss transform with a standard estimator (Johnson and Lindenstrauss, 1984) can be broken using only $O(k)$ queries. Similarly, (Ben-Eliezer et al., 2021b) gave a linear-size attack on the AMS sketch (Alon et al., 1999a), Cohen et al. (2022a) designed a linear-size attack against Count-Sketch (Charikar et al., 2002), and (Reviriego and Ting, 2020; Paterson and Raynal, 2021; Ahmadian and Cohen, 2024b) gave an efficient attack for MinHash cardinality sketches (Flajolet and Martin, 1985; Cohen, 1997; Flajolet et al., 2007; Heule et al., 2013).

We observe that an attack on any family of sketching maps that can be implemented in the query model and uses queries with non-negative values and with a constant (or logarithmic) ratio range of statistics values can be implemented in resettable streaming. The reduction is simple: move from one dataset $D_1 = \{(x, v_x)\}$ to $D_2 = \{(x, v'_x)\}$ by issuing the updates $\mathsf{Reset}(x); \mathsf{Inc}(x, v'_z))$ to each key in the support of $D_1 \cup D_2$. Then query for the (prefix-max) approximate value of the statistics. This transferability in particular holds for the attacks on MinHash sketches by Ahmadian and Cohen (2024a), and the attacks on union-composable cardinality sketches and on Real-valued linear $\ell_0$ sketches sketches by Cohen et al. (2024). Since resettable cardinality falls out as a special case for all frequency statistics (per Eq. (1)), this broadly suggests that union-composable or linear sketching maps fundamentally can not provide a positive answer to Question 1.1.

Intuitively, all sampling-based methods which utilize *persistent (sticky) per-key randomness* are vulnerable to adaptive attacks. This is important guidance since all known resettable streaming sketches are sampling-based (Gemulla et al., 2006; 2007; Cohen et al., 2012; Pavan et al., 2024; Lin et al., 2025). Sampling sketches (sometimes implicitly) are based on a sampling-to-order statistics transform where the value of each key is scaled by a random value and the sample is formed from order statistics (Knuth, 1998; Vitter, 1985; Rosén, 1997; Tillé, 2006; Cohen, 1997; Cohen and Kaplan, 2007; 2008). In composable (e.g. linear, MinHash) sketches, (Andoni et al., 2011; Jayaram and Woodruff, 2018; Cohen et al., 2020), the randomness is fixed for each key (sticky), which makes them inherently vulnerable with adaptive inputs.[10] Specifically, the resettable streaming sketches of (Lin et al., 2025) and known sampling sketches for super-linear statistics (including $\ell_2$ sampling) (Andoni et al., 2011; Jayaram and Woodruff, 2018; Cohen et al., 2020) utilize sticky per-key randomness and thus are vulnerable to the aforementioned adaptive attacks.

---

[9]Note that some attacks in the literature are designed against a specific estimator, while others are universal (apply for a sketch that uses any estimator).

[10]This is formalized in (Ahmadian and Cohen, 2024a; Cohen et al., 2024) as having a small *determining pool*.

# B. Analysis of the Fixed-Rate Robust cardinality with deletions

## B.1. Privacy layer (fixed-rate sketch)

Consider the fixed-rate cardinality sketch Algorithm 1 and recall that it maintains the (random) sample $S_t$ of size $S_t = |S_t|$ at time $t$ (we overload notation and use $S_t = |S_t|$ to also denote the sample size at time $t$ with the interpretation clear from context). Let

$$u_t := S_t - S_{t-1} \in \{-1, 0, +1\}$$

denote the (signed) change in sample size at time $t$ and observe that the prefix sum to step $t$ is the sample size at step $t$: $S_t = \sum_{t'=1}^{t} u_{t'}$. Recall that the standard estimator of the cardinality $N_t$ is the sample size scaled by $p$.

We feed the stream $(u_t)_{t=1}^{T}$ of sample-size updates into a Binary Tree Mechanism (Chan et al., 2011; Dwork et al., 2010; Dwork and Roth, 2014) (as described in Section G) with sensitivity parameter $L = 2$ and privacy parameter $\varepsilon_{\mathrm{dp}}$. The mechanism produces a sequence $(\tilde{S}_t)$ of noisy approximate prefix sums and our released estimates are

$$\hat{N}_t := \frac{\tilde{S}_t}{p}.$$

## B.2. Analysis: Error

We write the estimation error as

$$|\hat{N}_t - N_t| = \frac{1}{p} |\tilde{S}_t - pN_t|.$$

Then, we can decompose the error as follows:

$$|\tilde{S}_t - pN_t| \;\leq\; \underbrace{|\tilde{S}_t - S_t|}_{\text{(I) tree mechanism noise}} \;+\; \underbrace{|S_t - pN_t|}_{\text{(II) sampling fluctuation+adaptive bias}} \tag{7}$$

Term (I) is controlled by a uniform accuracy bound for the tree mechanism. Term (II) is controlled by a robust sampling lemma that exploits the stability guarantees of differential privacy.

### B.2.1. TERM (I): MECHANISM NOISE

The first term depends solely on the properties of the Tree Mechanism and is independent of the sampling or the adversary's strategy. Let $\varepsilon_{\mathrm{dp}}$ be the privacy parameter. From Lemma G.1 we have that for some absolute constant $C_1$, with probability $1 - \delta$, for all $t \in [T]$:

$$|\tilde{S}_t - S_t| \;\leq\; C_1 \frac{2}{\varepsilon_{\mathrm{dp}}} \log^{3/2} T \cdot \log \frac{T}{\delta}. \tag{8}$$

### B.2.2. TERMS (II): ROBUST SAMPLING

**Lemma B.1** (Robust sampling via DP generalization). *Let $(S_t)_{t=1}^{T}$ denote the sample-size process of Algorithm 1 with sampling rate $p \in (0,1)$, and let $N_t$ be the (random) number of active keys at time $t$. Let $N_{\max} := \max_{t \leq T} N_t$, and let $\varepsilon_{\mathrm{dp}} \in (0,1]$ be the privacy parameter of the tree mechanism applied to the sample-size updates.*

*Then there exists a universal constant $C > 0$ such that for any $\delta \in (0,1)$, with probability at least $1 - \delta$ over the sampling bits, the tree-mechanism noise, and any internal randomness of the adversary, we have simultaneously for all $t \in [T]$:*

$$\left| S_t - pN_t \right| \;\leq\; C \left( \varepsilon_{\mathrm{dp}}\, pN_t \;+\; \frac{1}{\varepsilon_{\mathrm{dp}}} \log \frac{T}{\delta} \right).$$

*Proof of Lemma B.1.* We apply the DP generalization theorem (Theorem F.1). We specify our sensitive units, their distribution, and functions that depend only on post-processed protected output (in particular, the input stream itself) so that the sample size can be expressed in the form of a linear sum of per-unit functions applied to unit values.

**Dataset and distribution.** Fix a time horizon $T$ and sampling rate $p \in (0, 1)$. Consider the (finite) set of *potential sampling units*

$$U := \{(x, j)\},$$

where $x$ ranges over all keys that are ever inserted in the stream, and $j \in \{1, 2, \dots\}$ indexes the *insertion intervals* of $x$: Each interval start in an insertion operation and ends just before the next operation (insertion or deletion) involving $x$. In each such interval it holds that either the key is in the sample in all time steps or it is not. By construction, the total number of such units is at most $T$ and each such units $u = (x, j) \in U$ corresponds to at most one distinct insertion event in the stream.

For each unit $u \in U$, we have an associated sampling bit $B_u \sim \text{Bernoulli}(p)$, sampled independently once and then fixed. Thus the dataset of sampling bits is

$$S = (B_u)_{u \in U} \in X^d, \qquad X = \{0, 1\}, \ d = |U| \le T,$$

and the underlying data distribution is $\mathcal{D} = \text{Bernoulli}(p)$.

**DP mechanism and post-processing.** We apply Lemma G.4. The contributions of each unit to the updates consist of at most one $+1$ contribution (if the key is sampled in when inserted) that is followed by a possible $-1$ contribution (if inserted, when re-inserted or deleted). The $L_1$ norm of the contributions is at most 2. Therefore, the outputs of the tree mechanism with $L = 2$ are unit-level $\varepsilon_{\text{dp}}$-DP with respect to $(B_u)_{u \in U}$.

The adversary observes the noisy prefixes $(\tilde{S}_t)_{t \le T}$ and chooses the update stream adaptively. By post-processing invariance of DP, the joint mapping

$$S \mapsto \left((\tilde{S}_t)_{t \le T}, \text{update stream}, (A_t)_{t \le T}\right)$$

is also $\varepsilon_{\text{dp}}$-DP with respect to $S$.

**Queries $h^{(t)}$.** For each time $t \in [T]$, and each unit $u = (x, j) \in U$, define the function $h_u^{(t)} : \{0, 1\} \to [0, 1]$ by

$$h_u^{(t)}(b) := \mathbf{1}\{u \text{ is active at time } t\} \cdot b.$$

Here "$u$ is active at time $t$" means that the key $x$ is currently active at time $t$ and this time lies in its $j$-th insertion interval. Each active key $x$ belongs to exactly one such unit $u = (x, j)$ at time $t$, so the number of active units is exactly $N_t$.

For a dataset $Y = (y_u)_{u \in U} \in \{0, 1\}^U$, define

$$h^{(t)}(Y) := \sum_{u \in U} h_u^{(t)}(y_u).$$

Note that for the actual sampling bits $S = (B_u)_{u \in U}$ we have

$$h^{(t)}(S) = \sum_{u \in U} \mathbf{1}\{u \text{ active at } t\} B_u = S_t,$$

the true sample size at time $t$.

Moreover, since $h_u^{(t)}$ depends on $S$ only through the event "$u$ is active at time $t$" (which is fully determined by the DP transcript and the adversary), and the distribution $\mathcal{D}$ is $\text{Bernoulli}(p)$ on each coordinate, we have

$$\mathop{\mathsf{E}}_{Z \sim \mathcal{D}^d}[h^{(t)}(Z)] = \sum_{u \in U} \mathbf{1}\{u \text{ active at } t\} \, \mathsf{E}[B'_u] = p \, N_t,$$

where $B'_u \sim \text{Bernoulli}(p)$ are fresh, independent copies.

Thus, for each fixed time $t$, $h^{(t)}$ is a query of the form covered by Theorem F.1, and we have

$$h^{(t)}(S) = S_t, \qquad \mathop{\mathsf{E}}_{Z \sim \mathcal{D}^d}[h^{(t)}(Z)] = pN_t.$$

**Applying DP generalization.** Fix a time $t \in [T]$ and apply Theorem F.1 with $\xi = 2\varepsilon_{\mathrm{dp}}$ and the query $h^{(t)}$. For any $\beta \in (0, 1]$, with probability at least $1 - \beta$ over the randomness of $S$ and the DP mechanism, we have

$$e^{-2\varepsilon_{\mathrm{dp}}} pN_t - S_t \leq \frac{2}{\varepsilon_{\mathrm{dp}}} \log\left(1 + \frac{1}{\beta}\right),$$

$$S_t - e^{2\varepsilon_{\mathrm{dp}}} pN_t \leq \frac{2}{\varepsilon_{\mathrm{dp}}} \log\left(1 + \frac{1}{\beta}\right).$$

Rewriting these as bounds on $|S_t - pN_t|$, we obtain

$$|S_t - pN_t| \;\leq\; (e^{2\varepsilon_{\mathrm{dp}}} - 1) pN_t \;+\; \frac{2}{\varepsilon_{\mathrm{dp}}} \log\left(1 + \frac{1}{\beta}\right).$$

Since $e^{4\varepsilon_{\mathrm{dp}}} - 1 \leq C'\varepsilon_{\mathrm{dp}}$ for $\varepsilon_{\mathrm{dp}} \leq 1$ and a universal constant $C'$, this implies

$$|S_t - pN_t| \;\leq\; C'\varepsilon_{\mathrm{dp}} pN_t \;+\; \frac{2}{\varepsilon_{\mathrm{dp}}} \log\left(1 + \frac{1}{\beta}\right).$$

**Uniform-in-time bound.** To obtain a bound that holds simultaneously for all $t \in [T]$, we apply a union bound. Set $\beta := \delta/T$. Then with probability at least $1 - \delta$,

$$|S_t - pN_t| \;\leq\; C'\varepsilon_{\mathrm{dp}} pN_t \;+\; \frac{4}{\varepsilon_{\mathrm{dp}}} \log\left(1 + \frac{T}{\delta}\right) \qquad \text{for all } t \in [T].$$

Absorbing constants and using $\log(1 + T/\delta) \leq C'' \log(T/\delta)$ for a universal constant $C'' > 0$ yields

$$|S_t - pN_t| \;\leq\; C\left(\varepsilon_{\mathrm{dp}} pN_t + \frac{1}{\varepsilon_{\mathrm{dp}}} \log\frac{T}{\delta}\right) \qquad \text{for all } t \in [T],$$

for some universal constant $C > 0$, as claimed. $\qquad\square$

### B.3. Proof for the fixed-rate sketch

We wrap up the analysis of the fixed-rate sketch.

*Proof of Theorem 2.4 with* (5). We apply the error decomposition (20) together with the accuracy of the tree mechanism (Term I, (8)) and the robust sampling bound (Lemma B.1), with constants scaled to yield the final specified guarantees.

With probability at least $1 - \delta$, simultaneously for all $t \in [T]$,

$$|\hat{N}_t - N_t| = \frac{1}{p}\left|\tilde{S}_t - pN_t\right|$$

$$\leq \frac{1}{p}\left(C_2 \frac{\log^{3/2} T}{\varepsilon_{\mathrm{dp}}} \log\frac{T}{\delta} \;+\; C\left(\varepsilon_{\mathrm{dp}} pN_{\max} + \frac{1}{\varepsilon_{\mathrm{dp}}} \log\frac{T}{\delta}\right)\right).$$

We now choose parameters. Set the DP noise parameter to match the target error:

$$\varepsilon_{\mathrm{dp}} := \Theta(\varepsilon).$$

This ensures that the bias term $\varepsilon_{\mathrm{dp}} pN_{\max}$ contributes at most $\Theta(\varepsilon pN_{\max})$.

Next choose the sampling rate

$$p \;:=\; \Theta\left(\frac{\log^{3/2} T \, \log(T/\delta)}{\varepsilon^2 N_{\max}}\right).$$

With this choice,

$$\frac{1}{p}\left(C_2 \frac{\log^{3/2} T}{\varepsilon_{\mathrm{dp}}} \log \frac{T}{\delta} + C \frac{1}{\varepsilon_{\mathrm{dp}}} \log \frac{T}{\delta}\right) \le \varepsilon N_{\max},$$

and the DP-bias term

$$\frac{1}{p}\,\varepsilon_{\mathrm{dp}}\, p N_{\max} \le \varepsilon N_{\max}.$$

Thus every error component is at most $\varepsilon N_{\max}$, giving (with appropriate per-term partitioning and scaling of $\delta$ and $\varepsilon$, absorbing into constants)

$$\max_{t \le T} |\hat{N}_t - N_t| \le \varepsilon N_{\max}.$$

**Space complexity.**  The sketch stores the sample $S_t$ together with $O(\log T)$ tree mechanism counters.

From Lemma B.1, with probability $1 - \delta$,

$$\max_{t \le T} S_t \le p N_{\max} + C\left(\varepsilon_{\mathrm{dp}}\, p N_{\max} + \frac{1}{\varepsilon_{\mathrm{dp}}} \log \frac{T}{\delta}\right).$$

Under our parameter choices, this becomes

$$\max_{t \le T} S_t = O\left(\frac{1}{\varepsilon^2} \log^{3/2} T \cdot \log \frac{T}{\delta}\right).$$

The $O(\log T)$ auxiliary state of the tree mechanism is asymptotically dominated by this term.

Thus the total space is

$$O\left(\frac{1}{\varepsilon^2} \log^{3/2} T \cdot \log \frac{T}{\delta}\right), \tag{9}$$

as claimed. $\square$

**Remark B.2.**  One can obtain the prefix-max guarantee (4) by running in parallel dyadic-rate instances of the fixed-rate robust sketch (Algorithm 1) with $p_i = 2^{-i}$. At each time, output the estimate from the *largest* rate (smallest $i$) whose instance has not been discarded. An instance is discarded once its *noisy* sample size exceeds a prescribed budget threshold. This generic "multi-rate" approach incurs an extra $O(|\mathcal{P}|)$ factor in space (typically $|\mathcal{P}| = O(\log T)$), since it maintains multiple sketches and DP trees. We next present a sketch with organic rate adjustment that avoids this overhead.

## C. Analysis of the adjustable-rate robust cardinality sketch

The robust adjustable-rate algorithm is given in Algorithm 2. We fix $L = 2$ for unit-level sensitivity, and set

$$k = \Theta\left(\frac{1}{\varepsilon^2} \log^{3/2} T \cdot \log \frac{T}{\delta}\right), \qquad \alpha = \Theta\left(\frac{1}{\varepsilon_{\mathrm{dp}}} \log^{3/2} T \cdot \log \frac{T}{\delta}\right), \qquad \varepsilon_{\mathrm{dp}} = \Theta(\varepsilon). \tag{10}$$

We show that with these parameter setting (with appropriate constants) Algorithm 2 satisfies Theorem 2.4 with the prefix-max error guarantees (4).

### C.1. Bounding the maximum sample size

By Lemma H.1, with probability at least $1 - \delta$ the algorithm performs at most $O(\log T)$ rate-halving steps over the entire execution. Hence the total number of calls to the tree mechanism is at most $T_{\mathrm{tree}} = T + O(\log T)$.

From the tree-accuracy bound Lemma G.1, with probability at least $1 - \delta$, it holds that for all calls $t \in [T_{\mathrm{tree}}]$

$$|\tilde{S}_t - S_t| \le C_1 \frac{L}{\varepsilon_{\mathrm{dp}}} \log^{3/2} T_{\mathrm{tree}} \cdot \log \frac{T_{\mathrm{tree}}}{\delta}.$$

All hidden constants are chosen so that $k \ge 4\alpha$ and the conditions of Lemma G.1 and Lemma C.2 hold.

**Claim C.1** (Sample size remains bounded by $k$)**.** For appropriate setting of constants in Eq. (10), with probability at least $1 - \delta$, for all $t \in [T]$, $S_t \leq k$.

*Proof.* Fix any time $t$. If $\tilde{S}_t \leq k - \alpha$, then by the tree accuracy bound $S_t \leq \tilde{S}_t + \alpha \leq k$. Otherwise, if $\tilde{S}_t > k - \alpha$, the algorithm repeatedly halves the sampling rate and subsamples until the loop condition fails, i.e., until $\tilde{S}_t \leq k - \alpha$, at which point the same argument applies. □

## C.2. Bounding the robust sampling error

The Bernoulli sampling randomness in Algorithm 2 can be alternatively specified by independent seeds $R_u \sim U[0, 1]$ associated with *units* $u = (x, j)$ of key $x$ and an insertion interval $j$: The key is included in the sample $S_t$ for $t$ in which the unit is active ($t$ lies in the $j$th insertion interval of $x$) if and only if $p_t \geq R_u$.

**Lemma C.2** (Multi-rate robust sampling)**.** *There exists a universal constant $C > 0$ such that for any $\delta \in (0, 1)$, with probability at least $1 - \delta$ over the seeds $(R_u)_u$, the randomness of the tree mechanism, and any internal randomness of the adversary, we have simultaneously for all $t \in [T]$:*

$$\left| S_t - p_t N_t \right| \leq C \left( \varepsilon_{\mathrm{dp}}\, p_t N_t + \frac{1}{\varepsilon_{\mathrm{dp}}} \log \frac{T}{\delta} \right).$$

*Proof.* Let $\mathcal{P} := \{p_0, p_0/2, p_0/4, \ldots, p_{\min}\}$ be the (random) set of sampling rates used by the algorithm, where $p_{\min}$ is the smallest rate reached with probability at least $1 - \delta$.

We index *units* by $u = (x, j)$ (key $x$ and insertion interval $j$) and associate with each unit an independent seed $R_u \sim U[0, 1]$. Let $S = (R_u)_{u \in U} \in [0, 1]^U$ denote the dataset of seeds.

For each time $t \in [T]$ and rate $p \in \mathcal{P}$, define the query

$$h_{t,p}(S) := \sum_{u \in U} \mathbf{1}\{u \text{ is active at time } t\}\, \mathbf{1}\{R_u \leq p\}. \tag{11}$$

Then $h_{t,p}(S) = S_t$ when $p = p_t$, and for all $(t, p)$,

$$\mathsf{E}[h_{t,p}(S)] = pN_t, \qquad 0 \leq h_{t,p}(S) \leq N_t.$$

Moreover, $h_{t,p}$ has Hamming sensitivity 1 with respect to $S$.

Let $\mathcal{M}$ denote the full interactive mechanism consisting of the tree mechanism and the adversary generating the update stream. By Lemma G.4 (with $L = 2$) and post-processing, $\mathcal{M}$ is $\varepsilon_{\mathrm{dp}}$-DP with respect to unit-level adjacency on $S$, even under fully adaptive updates.

Applying the DP generalization theorem (Theorem F.1) to each fixed query $h_{t,p}$ and taking a union bound over all $(t, p) \in [T] \times \mathcal{P}$, we obtain that with probability at least $1 - \delta$,

$$\left| h_{t,p}(S) - pN_t \right| \leq C \left( \varepsilon_{\mathrm{dp}}\, pN_t + \frac{1}{\varepsilon_{\mathrm{dp}}} \log \frac{T|\mathcal{P}|}{\delta} \right) \qquad \forall (t, p) \in [T] \times \mathcal{P}. \tag{12}$$

In particular, at each time $t$ the algorithm's actual sample size satisfies $S_t = h_{t,p_t}(S)$ for its current rate $p_t \in \mathcal{P}$, and the bound applies to $(t, p_t)$.

Finally, by Lemma H.1, with probability at least $1 - \delta$ the algorithm performs at most $O(\log T)$ rate halvings, and hence $|\mathcal{P}| = O(\log T)$. Substituting this into (12) yields the stated bound. □

## C.3. Relating sampling rate to cardinality

**Lemma C.3** (Rate at phase boundaries is $\Theta(k/\max_{t' \leq t} N_{t'})$)**.** *Run Algorithm 2 with sample budget $k$ and margin $\alpha$, and let $\mathcal{P} = \{p_0, p_0/2, p_0/4, \ldots\}$ be its dyadic rate grid. Let $0 = t_0 < t_1 < \cdots < t_m \leq T$ be the (random) times at which the algorithm performs a rate decrease (the "adjustments"); that is, at each $t_j$ the condition $\tilde{S}_{t_j} > k - \alpha$ triggers and the*

*algorithm updates $p \leftarrow p/2$ and subsamples the maintained set. Let $p^{(j)}$ denote the rate* after *the adjustment at time $t_j$ (with $p^{(0)} := p_0$), and let*

$$M_j := \max_{t \leq t_j} N_t$$

*be the prefix-max active cardinality up to the end of time $t_j$.*

*Assume the tree accuracy event holds:*

$$\forall t \in [T], \qquad |\tilde{S}_t - S_t| \leq \alpha, \tag{13}$$

*and assume the robust sampling bound holds uniformly for all $(t, p) \in [T] \times \mathcal{P}$:*

$$\forall (t, p) \in [T] \times \mathcal{P}, \qquad |S_{t,p} - pN_t| \leq C\left(\varepsilon_{\mathrm{dp}} \, pN_t + \frac{1}{\varepsilon_{\mathrm{dp}}} \log \frac{T|\mathcal{P}|}{\delta}\right), \tag{14}$$

*for a universal constant $C > 0$.*

*Then there exist absolute constants $0 < c < C' < \infty$ such that, on the event (13)–(14), for every adjustment index $j \in \{1, \ldots, m\}$ we have*

$$c \frac{k}{M_j} \leq p^{(j-1)} \leq C' \frac{k}{M_j}. \tag{15}$$

*In words:* the rate used immediately *before* an adjustment is within constant factors of $k$ divided by the current prefix maximum.

*Proof.* Fix an adjustment time $t_j$ and let $p := p^{(j-1)}$ be the rate in force immediately before the adjustment at time $t_j$.

**Lower bound on $p$.**  Since $t_j$ triggers an adjustment, we have $\tilde{S}_{t_j} > k - \alpha$. Using (13), the true sample size under rate $p$ satisfies

$$S_{t_j, p} = S_{t_j} \geq \tilde{S}_{t_j} - \alpha > k - 2\alpha.$$

Apply (14) to $(t_j, p)$:

$$S_{t_j, p} \leq p N_{t_j} + C\left(\varepsilon_{\mathrm{dp}} \, p N_{t_j} + \frac{1}{\varepsilon_{\mathrm{dp}}} \log \frac{T|\mathcal{P}|}{\delta}\right).$$

Rearranging (and taking $\varepsilon_{\mathrm{dp}}$ small enough so that $1 - C\varepsilon_{\mathrm{dp}} \geq 1/2$) yields

$$p N_{t_j} \geq \tfrac{1}{2} S_{t_j, p} - O\left(\frac{1}{\varepsilon_{\mathrm{dp}}} \log \frac{T}{\delta}\right).$$

With the chosen parameters ($k \gg \frac{1}{\varepsilon_{\mathrm{dp}}} \log(T/\delta)$ and $\alpha = o(k)$), the right-hand side is $\Omega(k)$, and hence $p N_{t_j} \geq \Omega(k)$. Since $N_{t_j} \leq M_j$, this implies

$$p \geq c \frac{k}{M_j}$$

for a suitable absolute constant $c > 0$.

**Upper bound on $p$.**  Consider the moment *right after* the adjustment at time $t_j$. The algorithm updates the sampling rate $p \leftarrow p/2$ (so the new rate is $p^{(j)} = p/2$) and subsamples the maintained set accordingly. It then issues the bulk update to the tree mechanism and obtains a refreshed reported value $\tilde{S}_{t_j}^{\mathrm{post}}$. By definition of the while-loop trigger, the algorithm exits the adjustment step only once $\tilde{S}_{t_j}^{\mathrm{post}} \leq k - \alpha$. On the event (13), this implies that the corresponding true sample size at rate $p^{(j)}$ satisfies $S_{t_j, p^{(j)}} \leq k$ (absorbing constant slack into $k$).

Applying (14) to $(t_j, p^{(j)})$ gives

$$p^{(j)} N_{t_j} \leq S_{t_j, p^{(j)}} + C\left(\varepsilon_{\mathrm{dp}} \, p^{(j)} N_{t_j} + \frac{1}{\varepsilon_{\mathrm{dp}}} \log \frac{T|\mathcal{P}|}{\delta}\right) \leq k + o(k),$$

and hence $p^{(j)} N_{t_j} \leq O(k)$. Since $p^{(j)} = p/2$, this implies $p \leq O(k/N_{t_j})$.

Finally, because rates are dyadic and an adjustment is triggered only when $pN_{t_j} = \Omega(k)$, the active cardinality at the trigger time must satisfy $N_{t_j} = \Theta(M_j)$ up to constant factors; otherwise the dyadic rate preceding the adjustment would already violate the trigger condition. Since always $N_{t_j} \leq M_j$, we conclude

$$p \leq C' \frac{k}{M_j}$$

after adjusting constants. This completes the proof of (15).

$\square$

**Corollary C.4** (Rate within constant factors of $k/\max_{t' \leq t} N_{t'}$)**.** Let the setup be as in Lemma C.3. For $j \in \{0, 1, \ldots, m\}$ define the $j$-th *phase* to be the time interval

$$\{t : t_j < t \leq t_{j+1}\}, \qquad \text{where we set } t_{m+1} := T.$$

(Thus the rate is constant throughout each phase.) Let $p_t$ denote the rate used by the algorithm at time $t$ and let $M_t := \max_{t' \leq t} N_{t'}$.

On the event (13)–(14), there exist absolute constants $0 < c < C < \infty$ such that for all $t \in [T]$,

$$c\,\frac{k}{M_t} \ \leq \ p_t \ \leq \ C\,\frac{k}{M_t}. \tag{16}$$

Consequently,

$$p_t \ = \ \Theta\!\left(\frac{k}{\max_{t' \leq t} N_{t'}}\right) \ = \ \Theta\!\left(\frac{\log^{3/2} T \, \log(T/\delta)}{\varepsilon^2 \max_{t' \leq t} N_{t'}}\right),$$

after substituting the choice $k = \Theta(\varepsilon^{-2} \log^{3/2} T \log(T/\delta))$.

*Proof.* Fix $t \in [T]$ and let $j$ be the (unique) phase index such that $t_j < t \leq t_{j+1}$. By definition of the algorithm, the sampling rate is constant during phase $j$, hence $p_t = p^{(j)}$.

Apply Lemma C.3 to the *next* adjustment boundary $t_{j+1}$ (if $j = m$, interpret $t_{m+1} = T$ and note that no further decrease occurs, so the same upper/lower bounds hold with $M_{m+1} := \max_{t' \leq T} N_{t'}$). Lemma C.3 gives constants $c_0, C_0$ such that

$$c_0\,\frac{k}{M_{j+1}} \ \leq \ p^{(j)} \ \leq \ C_0\,\frac{k}{M_{j+1}},$$

where $M_{j+1} = \max_{t' \leq t_{j+1}} N_{t'}$.

Since $t \leq t_{j+1}$, we have $M_t \leq M_{j+1}$. Moreover, the prefix maximum cannot grow by more than a constant factor within a phase without triggering an adjustment; hence $M_{j+1} \leq 2M_t$. Substituting $M_{j+1} \in [M_t, 2M_t]$ into the bound on $p^{(j)}$ yields (16) with $c = c_0/2$ and $C = 2C_0$. $\square$

### C.4. Wrapping up the proof

*Proof of Theorem 2.4 with* (4)*.* We analyze Algorithm 2 under the parameter choice (10). Let $\mathcal{E}$ be the event that (i) the tree-mechanism accuracy bound holds uniformly, i.e. $|\tilde{S}_t - S_t| \leq \alpha$ for all $t \in [T]$, and (ii) the multi-rate robust sampling bound of Lemma C.2 holds uniformly for all $t \in [T]$. By allocating failure probability $\delta/2$ to each event and applying a union bound, Lemma G.1 and Lemma C.2 imply $\Pr[\mathcal{E}] \geq 1 - \delta$. We condition on $\mathcal{E}$ for the rest of the analysis.

**Space bound.** By Claim C.1, on $\mathcal{E}$ we have $S_t \leq k$ for all $t \in [T]$. The sketch stores the current sample set (size at most $k$) plus $O(\log T_{\text{tree}}) = O(\log T)$ counters for the tree mechanism, so the total space is $O(k)$, i.e. $O\!\left(\varepsilon^{-2} \log^{3/2} T \cdot \log(T/\delta)\right)$.

**Error decomposition.** Fix a time $t \in [T]$. The released estimate is $\hat{N}_t = \tilde{S}_t / p_t$, where $p_t$ is the current sampling rate at time $t$. Write the estimation error as

$$|\hat{N}_t - N_t| = \frac{1}{p_t} |\tilde{S}_t - p_t N_t| \leq \frac{1}{p_t} \left( |\tilde{S}_t - S_t| + |S_t - p_t N_t| \right),$$

where $S_t$ denotes the true (random) sample size at time $t$. On $\mathcal{E}$, the tree term is bounded by $|\tilde{S}_t - S_t| \leq \alpha$, and Lemma C.2 gives

$$|S_t - p_t N_t| \leq C \left( \varepsilon_{\mathrm{dp}} \, p_t N_t + \frac{1}{\varepsilon_{\mathrm{dp}}} \log \frac{T}{\delta} \right).$$

Substituting,

$$|\hat{N}_t - N_t| \leq \underbrace{\frac{\alpha}{p_t}}_{\text{tree noise}} + \underbrace{C \varepsilon_{\mathrm{dp}} N_t}_{\text{DP bias}} + \underbrace{\frac{C}{\varepsilon_{\mathrm{dp}} \, p_t} \log \frac{T}{\delta}}_{\text{generalization term}} . \tag{17}$$

**Relating $p_t$ to the prefix maximum.** Let $M_t := \max_{t' \leq t} N_{t'}$. By Corollary C.4, on $\mathcal{E}$ the current rate satisfies

$$p_t = \Theta\left( \frac{k}{M_t} \right), \qquad \text{equivalently} \qquad \frac{1}{p_t} = \Theta\left( \frac{M_t}{k} \right). \tag{18}$$

**Finishing the bound.** Plug (18) into (17). First,

$$\frac{\alpha}{p_t} = \Theta\left( \alpha \cdot \frac{M_t}{k} \right).$$

With $\alpha = \Theta\left( \varepsilon_{\mathrm{dp}}^{-1} \log^{3/2} T \log(T/\delta) \right)$ and $k = \Theta\left( \varepsilon^{-2} \log^{3/2} T \log(T/\delta) \right)$, and $\varepsilon_{\mathrm{dp}} = \Theta(\varepsilon)$, this becomes $\Theta(\varepsilon M_t)$.

Second,

$$\frac{1}{\varepsilon_{\mathrm{dp}} \, p_t} \log \frac{T}{\delta} = \Theta\left( \frac{M_t}{\varepsilon_{\mathrm{dp}} \, k} \log \frac{T}{\delta} \right) = O(\varepsilon M_t),$$

since $k$ contains the factor $\log(T/\delta)$ and $\varepsilon_{\mathrm{dp}} = \Theta(\varepsilon)$.

Finally, the middle term in (17) satisfies $C \varepsilon_{\mathrm{dp}} N_t \leq C \varepsilon_{\mathrm{dp}} M_t = O(\varepsilon M_t)$.

Combining these bounds and absorbing constants into the choice of parameters in (10), we obtain that for all $t \in [T]$,

$$|\hat{N}_t - N_t| \leq \varepsilon M_t = \varepsilon \max_{t' \leq t} N_{t'}.$$

This proves the prefix-max error guarantee (4) on the event $\mathcal{E}$, which holds with probability at least $1 - \delta$. $\qquad \square$

## D. Robust Resettable Sum Analysis

As a stepping stone to the full algorithm which satisfies the prefix-max error guarantee, we first show that a slightly-simplified version of our algorithm achieves additive error $\varepsilon \cdot \max_{t \in [T]} F_t$.

**Lemma D.1.** *Given accuracy parameter $\varepsilon > 0$ and failure probability $\delta > 0$, there exists an adversarially robust sketch for the sum $F = \sum_x v_x$ under a stream of $T$ insertions and reset operations that, with probability at least $1 - \delta$, outputs an estimate $\hat{F}_t$ such that*

$$|\hat{F}_t - F_t| \leq \varepsilon \cdot \max_{t \in [T]} F_t = \varepsilon \cdot F_{\max} \quad \text{for all } t \in [T] \tag{19}$$

*The algorithms uses*

$$O\left( \frac{1}{\varepsilon^2} \log^{9/2}(T) \cdot \log^2 \left( \frac{1}{\delta} \right) \right)$$

*bits of memory.*

Then, in Section D.5 we use sketch switching to obtain our final algorithm, which satisfies the prefix-max error guarantee $\varepsilon \cdot \max_{t' \leq t} F_t$ as in Theorem 3.1.

## D.1. Standard Sketch for Resettable Sum

We begin by analyzing the properties of the standard non-robust sketch, framing it in a way that facilitates the robust extension in subsequent sections.

The standard (non-robust) resettable sum algorithm is described in Algorithm 3. This is a specialization of the ReLU sketch of Cohen et al. (2012) (with a unit update ReLU version first proposed in (Gemulla et al., 2006; 2007)). The sampling threshold $\tau$ plays the role of an inverse sampling rate. The sketch applies "sample-and-hold" Gibbons and Matias (1998); Estan and Varghese (2002), where updates to keys that are in the sample are implemented by updating the stored count.

### D.1.1. PROPERTIES FOR NON-ADAPTIVE STREAMS

We will work with an equivalent view of Algorithm 3 that casts it as a deterministic function of the input stream and a set of fixed, independent random variables (entry thresholds), with independent contributions to the estimate and sample.

**Definition D.2** (Entry-Threshold Formulation). For each key $x$ and each reset epoch $j$, let $R_{x,j} \sim \mathrm{Exp}(1/\tau)$ be an independent *entry threshold*. Let $v_x^{(t)}$ denote the active value of key $x$ at time $t$ (accumulated since the last reset). The state of the sketch at time $t$ is determined strictly by the pairs $\{(v_x^{(t)}, R_{x,j})\}$:

1. **Sample Indicators:** The presence of key $x$ in the sample is given by the indicator variable $I_x^{(t)}$. Key $x$ is sampled if and only if its value exceeds its active threshold:

$$I_x^{(t)} \ := \ \mathbb{I}\left\{v_x^{(t)} > R_{x,j}\right\}.$$

   The total sample size is the sum of these independent indicators:

$$|S_t| \ = \ \sum_x I_x^{(t)}.$$

2. **Sum Estimator:** The contribution of key $x$ to the sum estimate is defined as:

$$Z_x^{(t)} \ := \ I_x^{(t)} \cdot \left(v_x^{(t)} + \tau - R_{x,j}\right).$$

   The total estimate is the sum of these independent contributions:

$$\hat{F}_t \ = \ \sum_x Z_x^{(t)}.$$

In this formulation of the algorithm, both the sample size and the estimate are expressed as sums of mutually independent random variables (conditioned on the input stream). In particular, the contribution of unit $(x, j, R_{x,j})$ depends only on its own value and the threshold $R_{x,j}$, not on any other unit.

**Concentration Properties.** We now derive the accuracy and space bounds. To prepare for the robust setting (where sensitivity must be bounded), we analyze the algorithm conditioned on the high-probability event that the thresholds are bounded.

**Lemma D.3** (Bounded Threshold Event). *Let $T$ be the stream length and $\delta \in (0, 1)$. Define $L := \tau \log(T/\delta)$. Let $\mathcal{E}$ be the event that for all keys $x$ and epochs $j$ active at any time $t \leq T$, the entry threshold satisfies $R_{x,j} \leq L$. Then $\Pr[\mathcal{E}] \geq 1 - \delta$.*

*Proof.* For a single exponential variable $R \sim \mathrm{Exp}(1/\tau)$, $\Pr[R > L] = e^{-L/\tau} = \delta/T$. Taking a union bound over at most $T$ active epochs implies the claim. $\square$

**Remark D.4** (Stability of Heavy Keys). Conditioned on the event $\mathcal{E}$, the algorithm exhibits a form of *stability* for large values. Specifically, for any key with value $v_x > L$, the sampling indicator becomes deterministic ($I_x = 1$) because $R_{x,j} \leq L < v_x$. Consequently, for any $v_x$, the dependence of the estimate on the specific value of $R_{x,j}$ is bounded by $L$ since $v_x \geq L \implies Z_x := v_x + \tau - R_{x,j} \in [v_x + \tau - L, v_x + \tau]$ and for $v_x \leq L$, $Z_x \in [0, L]$. This bounded sensitivity property is key in the analysis of a robust version.

**Lemma D.5** (Conditional Concentration). *Conditioned on the event $\mathcal{E}$ (i.e., assuming $R_{x,j} \leq L$ for all active $x$), the following bounds hold for any fixed time $t$ with probability at least $1 - \gamma$:*

1. *__Accuracy:__ The estimation error is bounded by:*

$$|\hat{F}_t - F_t| \leq \sqrt{2\tau F_t \log \tfrac{2}{\gamma}} + \frac{2}{3} L \log \tfrac{2}{\gamma}.$$

2. *__Space:__ The sample size $|S_t|$ is bounded by:*

$$|S_t| \leq \frac{F_t}{\tau} + \sqrt{\frac{2F_t}{\tau} \log \tfrac{1}{\gamma}} + \frac{2}{3} \log \tfrac{1}{\gamma}.$$

*Proof.* **Accuracy:** Let $E_x = Z_x - v_x$ be the error contribution of key $x$. We analyze the range of $E_x$ given $R_{x,j} \leq L$:

- If $v_x > L$: Then $I_x = 1$ is deterministic. The estimator is $Z_x = v_x + \tau - R_{x,j}$. The error is $E_x = \tau - R_{x,j}$. Since $0 \leq R_{x,j} \leq L$, we have $|E_x| \leq \max(\tau, L)$.

- If $v_x \leq L$:
  - If sampled ($I_x = 1$): $E_x = \tau - R_{x,j}$, so $|E_x| \leq L$.
  - If not sampled ($I_x = 0$): $Z_x = 0$, so $E_x = -v_x$. Since $v_x \leq L$, $|E_x| \leq L$.

Thus, conditioned on $\mathcal{E}$, the variable $Z_x$ has a deviation range bounded by $L$. Using the variance bound $\sum \mathsf{Var}[Z_x] \leq \tau F_t$ (from standard properties) and Bernstein's inequality yields the result.

**Space:** The sample size is $|S_t| = \sum_x I_x$. This is a sum of independent Bernoulli trials. Under $\mathcal{E}$, if $v_x > L$, then $I_x = 1$ deterministically (variance is 0 for these terms). For $v_x \leq L$, $I_x$ remains random. In all cases, the terms are independent and bounded by 1. The mean is $\mu = \sum (1 - e^{-v_x/\tau}) \leq F_t/\tau$. Bernstein's inequality yields the bound. $\square$

**Corollary D.6** (Summary of Trade-offs). Fix accuracy $\varepsilon$ and failure probability $\delta$. Set $\tau = \Theta(\varepsilon^2 F_{\max} / \log^2(T/\delta))$. With probability $1 - \delta$, the algorithm maintains a sketch of size $O(\varepsilon^{-2} \log(T/\delta))$ and provides an estimate $\hat{F}_t$ satisfying:

$$|\hat{F}_t - F_t| \leq \varepsilon F_{\max}.$$

In the analysis of our robust algorithm, we will further need to bound the error contribution of the keys $x$ with $v_x > L$. Let $D_t = \sum_x \max(v_x - L, 0)$ and $\hat{D}_t = \sum_x \max(v_x + \tau - R_{x,j} - L, 0)$.

**Corollary D.7** (Error of deterministic parts). Fix accuracy $\varepsilon > 0$ and failure probability $\delta > 0$. Let $\tau = O\left(\varepsilon^2 M / \log^2(T/\delta)\right)$. Then, with probability $1 - \delta$, we have $|\hat{D}_t - D_t| \leq \varepsilon\sqrt{M \cdot F_t} + \varepsilon^2 M$ at all $t \in [T]$.

*Proof.* This follows by almost the same argument as above.

Consider $x$ such that $v_x > L$. Let $Y_x = v_x + \tau - R_{x,j} - L$. The estimated contribution of key $x$ is $Z_x = \max(Y_x, 0)$, so the error is precisely $E_x = \tau - R_{x,j}$ when $Y_x > 0$. Otherwise, if $Y_x < 0$, the error is $E_x = v_x - L$. Now, recall that under event $\mathcal{E}$, we have that $0 \leq R_{x,j} \leq L$. Since we assumed that $Y_x = v_x + \tau - R_{x,j} - L < 0$, we have that $E_x \leq R_{x,j} - \tau$. Thus, it follows that $|E_x| \leq \max(\tau, L)$ in both cases. Since there can be at most $F_t/L$ keys $x$ with $v_x > L$ and $\mathsf{Var}[Z_x] \leq \tau^2$, it follows that $\sum_{x:v_x > L} \mathsf{Var}[Z_x] \leq \tau F_t$. Finally, for keys $x$ satisfying $v_x \leq L$, we have $Y_x = v_x + \tau - R_{x,j} - L \leq \tau - R_{x,j}$, so $|E_x| \leq \max(\tau, L)$ and $\mathsf{Var}[Z_x] \leq \tau v_x$.

By standard properties, we again conclude that $\sum_x \mathsf{Var}[Z_x] \leq 2\tau \cdot F_t$, and applying Bernstein's inequality yields

$$|\hat{D}_t - D_t| \leq \sqrt{2\tau F_t \log \tfrac{2}{\gamma}} + \frac{2}{3} L \log \tfrac{2}{\gamma}$$

Then, by setting $\gamma = \delta/T$, we can union bounding over all $T$ time steps, and obtain

$$|\hat{D}_t - D_t| \leq \sqrt{2\tau F_t \log \tfrac{T}{\delta}} + \frac{2}{3} L \log \tfrac{T}{\delta}$$

Finally, since $L = \tau \log(T/\delta)$, for $\tau$ chosen as above, we have

$$|\hat{D}_t - D_t| \leq \varepsilon \cdot \sqrt{M \cdot F_t} + \varepsilon^2 M$$

$\square$

### D.2. Privacy Layer

To obtain robustness, we compose Algorithm 3 with the Binary Tree Mechanism described in Section G, applied on a stream of scaled differences of the algorithm's estimates.

Let $F^{(t)} = \sum_x v_x^{(t)}$ be the value of the statistic at time $t$. Let $S_t$ denote the sample at time $t$, and for each $x \in S_t$ let $c_x^{(t)}$ be its associated counter.

**Corollary D.8.** Fix a time horizon $T$. Let $B := \tau \log(T/\delta)$. With probability $1 - \delta$, it holds that for all keys $x$ and $t \in [T]$, $c_x^{(t)} \geq v_x^{(t)} - B$.

In fact, it holds that for all $x$ and $t \in [T]$ we have $c_x^{(t)} \geq v_x^{(t)} - B$ if and only if $R_{x,j} \leq B$ for all applicable units $(x, j)$. So, Corollary D.8 follows from Lemma D.3.

We condition on the event

$$\mathcal{E} := \{R_{x,j} \leq B \text{ for all applicable } (x, j)\}$$

which holds with probability $1 - \delta$ by Corollary D.8.

Now, we separate the contribution of each key $x$ to $F^{(t)}$ into the *deterministic part* $\max(0, v_x^{(t)} - B)$ which is known to the adversary, and the *uncertain part* $\min(B, v_x^{(t)})$ which will be hidden via the binary tree mechanism. Thus, at each time $t \in [T]$ we can decompose the total value of $F_t$ as follows:

$$F_t = \sum_x \max(0, v_x^{(t)} - B) + \sum_x \min(B, v_x^{(t)}) = D_t + P_t.$$

Recall that the non-private estimate returned by Algorithm 3 at time $t$ is

$$\hat{F}_t := \sum_{x \in S^{(t)}} \left(\tau + c_x^{(t)}\right) = \sum_{x \in S^{(t)}} \left(v_x^{(t)} - R_{x,j_x} + \tau\right).$$

Likewise, we decompose the contribution of each key $x \in S_t$ to the estimate $\hat{F}_t$:

$$\hat{F}_t = \sum_{x \in S_t} \max(0, v_x^{(t)} - R_{x,j_x} + \tau - B) + \sum_{x \in S_t} \min(B, v_x^{(t)} - R_{x,j_x} + \tau) = \hat{D}_t + \hat{P}_t$$

We define the update stream given to the tree mechanism by

$$u_t := \frac{1}{B}\left(\hat{P}_t - \hat{P}_{t-1}\right).$$

We further partition updates $\{u_t\}_{t=1}^T$ based on the updated key $x$: for any $t \in [T]$, let $U_{x,j_x}^{(t)}$ denote the set of updates to coordinate $x$ after the $(j_x - 1)$-th reset and before time $t$. For each $x$ and $j_x > 1$, let the sequence of updates $\{u : u \in U_{x,j_x}^{(t)}\}$ be a single privacy unit. Importantly, since the input stream to the algorithm only contains element insertions and resets, it

follows that all positive updates $u$ satisfy $\sum_{u \in U^{(t)}_{x,j_x}} u \leq 1$ and there can be at most one negative update $u' = -\sum_{u \in U^{(t)}_{x,j_x}} u$. Thus, we have

$$\sum_{u \in U^{(t)}_{x,j_x}} |u| \leq 2.$$

We feed the stream of updates $(u_t)^T_{t=1}$ to the uncertain part $\hat{P}_t$ as input to the Binary Tree Mechanism with sensitivity $L = 2$ and privacy parameter $\varepsilon_{\mathrm{dp}}$. Let $\tilde{U}_t$ denote the noisy estimates of the prefix sums $\hat{U}_t = \frac{1}{B} \cdot \hat{P}_t$ produced by the tree mechanism on the stream $\{u_t\}^T_{t=1}$. Then, the estimate returned at time $t$ by the composed algorithm is

$$\tilde{F}_t := B\,\tilde{U}_t + \hat{D}_t.$$

## D.3. Analysis

We decompose the error of the adversarially robust algorithm as follows:

$$|\tilde{F}_t - F_t| \;\leq\; \underbrace{|\tilde{F}_t - \hat{F}_t|}_{\text{(I) DP mechanism noise}} \;+\; \underbrace{|\hat{F}_t - F_t|}_{\text{(II) sampling fluctuation+adaptive bias}} \tag{20}$$

In the next subsections, we show how to separately bound the contribution of the error from (I) the Binary Tree Mechanism and (II) the error of the algorithm's sampling under adaptive updates.

### D.3.1. TERM (I): MECHANISM NOISE

The first term (I) only depends on the properties of the Tree Mechanism, and in particular it is independent of the internal randomness of the algorithm and the adversarial strategy. By construction, each privacy unit $\{u : u \in U^{(t)}_{x,j_x}\}$ satisfies

$$\sum_{u \in U^{(t)}_{x,j_x}} |u| \leq 2$$

By Lemma G.4, with probability at least $1 - \delta$, we have $|\tilde{U}_t - U_t| \;\leq\; C_1 \frac{\log^{3/2} T}{\varepsilon_{\mathrm{dp}}} \cdot \log \frac{T}{\delta}$ and therefore

$$|\tilde{F}_t - \hat{F}_t| = |B \cdot \tilde{U}_t - B \cdot \hat{U}_t| \leq C_1 \cdot B \cdot \frac{\log^{3/2} T}{\varepsilon_{\mathrm{dp}}} \cdot \log \frac{T}{\delta}. \tag{21}$$

### D.3.2. TERM (II): ADAPTIVE BIAS

Next, we bound the sampling error and adaptive bias $|\hat{F}_t - F_t|$. To this end, we first show the following lemma, which gives an upper bound on $|\hat{P}_t - P_t|$.

**Lemma D.9** (Robust sampling via DP generalization). *Let $(\hat{P}^{(t)})^T_{t=1}$ denote the sequence of protected components of the estimate $\hat{F}^{(t)}$ produced by Algorithm 3 with sampling threshold $\tau$, and let $P^{(t)}$ be the true contribution of the protected part of $F^{(t)}$ at time $t$. Let $\varepsilon_{dp} \in (0,1]$ be the privacy parameter for the tree mechanism applied to updates $(u_t)^T_{t=1}$ defined above.*

*Then, there exists a universal constant $C > 0$ such that for any $\delta > 0$, with probability at least $1 - \delta$ over the randomness of the algorithm, the tree mechanism noise, and the internal randomness of the adversary, we have that for all $t \in [T]$ simultaneously,*

$$\left| \hat{P}^{(t)} - P^{(t)} \right| \leq C \cdot \left( \frac{B}{\varepsilon_{dp}} \log \left( 1 + \frac{T}{\delta} \right) + \varepsilon_{dp} \cdot F_{\max} \right)$$

*Proof.* We will apply the DP generalization theorem (Theorem F.1). To do so, we first define the sensitive units, their distribution, and predicates that depend only on the protected units of the tree mechanism $U^{(t)}_{x,j_x}$, which consist of sequences of updates to the private portion of each key $x$.

**Dataset and distribution.** Fix the time horizon $T$ and sampling threshold $\tau$. By Definition D.2, the algorithm can be viewed as first drawing an infinite sequence of i.i.d. entry thresholds $R_{x,1}, R_{x,2}, ... \sim \mathrm{Exp}(1/\tau)$ and then using threshold $R_{x,j}$ for the $j$-th re-insertion of $x$. Recall that at a given time $t$ during the $j$-th insertion interval, $x \in S^{(t)}$ if and only if $v_x^{(t)} > R_{x,j}$. Consider the set of potential sampling units

$$U := \{(x, j, R_{x,j})\}$$

where $x$ is any key, $j \in \{1, 2, ...\}$ represents the number of times that $x$ has been re-inserted into the stream following a reset operation, and $R_{x,j} \sim \mathrm{Exp}(1/\tau)$ is the corresponding entry threshold.

Define the dataset

$$S = \{u_1, ..., u_d\}, \quad u_i = (x_i, j_i, R_{x_i, j_i})$$

Under event $\mathcal{E}$, the distribution $\mathcal{D}$ for each entry threshold $R_{x,j}$ is $\mathrm{Exp}(1/\tau)$ conditioned on $R_{x,j} \le B$.

**DP mechanism.** By Lemma G.4, the algorithm producing outputs $\hat{F}^{(t)}$ is $\varepsilon_{\mathrm{dp}}$ with respect to event-level adjacency on the privacy units $U_{x,j}^{(t)}$, and the contributions of these units to $\hat{F}^{(t)}$ are determined by the value of $R_{x,j}$. Thus, it follows that the mechanism is also $\varepsilon_{\mathrm{dp}}$-DP with respect to the redefined privacy units $u = (x, j, R_{x,j}) \in S$.

**Defining Queries.** For each time $t \in [T]$ and unit $u = (x, j, R_{x,j}) \in U$, define the function

$$h_u^{(t)}(r) := \mathbf{1}\{u \text{ is active at time } t\} \cdot \left(\frac{1}{B} \min(B, v_x^{(t)} - r + \tau)\right), \quad r \in [0, B]$$

Observe that $h_u^{(t)}(r) : [0, B] \to [0, 1]$. For a dataset $Y = (y_u)_{u \in U}$, define

$$h^{(t)}(Y) := \sum_{u \in U} h_u^{(t)}(y_u).$$

For the actual dataset $S = (u_1, ..., u_d)$, we have that

$$h^{(t)}(S) = \sum_{u \in U} h_u^{(t)}(R_{x,j}) = \frac{1}{B} \cdot \hat{P}^{(t)}.$$

Next, draw $Z = (Z_1, ..., Z_d) \sim \mathcal{D}^d$ from the same conditional exponential distribution. We have that

$$\mathbb{E}_{Z \sim \mathcal{D}^d}[h^{(t)}(Z)] = \frac{1}{B} \sum_{u \in U} \mathbf{1}\{u \text{ is active at time } t\} \cdot \mathbb{E}_{Z \sim \mathcal{D}^d}[\min(B, v_x^{(t)} - Z_u + \tau)]$$

Since $\mathbb{E}[v_x^{(t)} - Z_u + \tau] = v_x^{(t)}$, and $v_x^{(t)} = \max(0, v_x^{(t)} - B) + \min(B, v_x^{(t)})$, it follows that

$$\mathbb{E}_{Z \sim \mathcal{D}^d}[h^{(t)}(Z)] = \frac{1}{B} \cdot P^{(t)}$$

**Applying DP generalization.** Fix time $t \in [T]$. Let $\xi = 2\varepsilon_{\mathrm{dp}}$. By Theorem F.1, for any $\beta \in (0, 1]$, with probability at least $1 - \beta$ over the randomness $S$ of the algorithm and the DP tree mechanism, we have that

$$\frac{1}{B} \left| e^{-2\varepsilon_{\mathrm{dp}}} \cdot P_t - \hat{P}_t \right| \le \frac{2}{\varepsilon_{\mathrm{dp}}} \log\left(1 + \frac{1}{\beta}\right)$$

By rearranging the inequality above, we have

$$\left|\hat{P}_t - P_t\right| \le \frac{2B}{\varepsilon_{\mathrm{dp}}} \log\left(1 + \frac{1}{\beta}\right) + (e^{-2\varepsilon_{\mathrm{dp}}} - 1) \cdot P_t$$

Observe that $e^{-2\varepsilon_{\mathrm{dp}}} - 1 \le C' \varepsilon_{\mathrm{dp}}$ for $\varepsilon_{\mathrm{dp}} \le 1$ for some universal constant $C'$, so

$$\left|\hat{P}_t - P_t\right| \le \frac{2B}{\varepsilon_{\mathrm{dp}}} \log\left(1 + \frac{1}{\beta}\right) + C' \varepsilon_{\mathrm{dp}} \cdot P_t$$

**Uniform-in-time bound.** Set $\beta := \delta/T$. By a union bound, with probability at least $1 - \delta$, we obtain

$$|\hat{P}_t - P_t| \le \frac{2B}{\varepsilon_{\mathrm{dp}}} \log\left(1 + \frac{T}{\delta}\right) + C' \varepsilon_{\mathrm{dp}} \cdot P_t, \quad \text{for all } t \in [T].$$

Since $\log\left(1 + T/\delta\right) \le C'' \cdot \log(T/\delta)$ for some constant $C'' > 0$, we have that

$$|\hat{P}_t - P_t| \le C \cdot \left(\frac{B}{\varepsilon_{\mathrm{dp}}} \log\left(1 + \frac{T}{\delta}\right) + \varepsilon_{\mathrm{dp}} \cdot P_t\right), \quad \text{for all } t \in [T] \tag{22}$$

for some universal constant $C > 0$. Since $\max_{t \in [T]} P_t \le \max_{t \in [T]} F_t = F_{\max}$, the claim follows.

$\square$

Finally, to bound the term $|\hat{F}_t - F_t|$ from Eq. (20), we recall that we decomposed $\hat{F}_t = \hat{D}_t + \hat{P}_t$ and similarly $F_t = D_t + P_t$. So, the error is bounded as follows:

$$\left|\hat{F}_t - F_t\right| \le \left|\hat{D}_t - D_t\right| + \left|\hat{P}_t - P_t\right|$$

Also, note that by applying Corollary D.7 with error parameter $\varepsilon'$, we have $|\hat{D}_t - D_t| \le \varepsilon' F_{\max}$. By combining the guarantees of the DP generalization analysis above with this, we obtain

$$\left|\hat{F}_t - F_t\right| \le \varepsilon' F_{\max} + C \cdot \left(\frac{B}{\varepsilon_{\mathrm{dp}}} \log\left(1 + \frac{T}{\delta}\right) + \varepsilon_{\mathrm{dp}} \cdot F_{\max}\right) \tag{23}$$

### D.4. Proof for fixed-rate sketch

In this subsection, we complete the analysis of the fixed-rate sketch for the sum $F = \sum_x v_x$ under adaptive increments and resets.

*Proof of Lemma D.1.* We recall the error decomposition in Eq. (20). In particular, we will apply the guarantee of the tree mechanism (Eq. (21)) and the robust sampling bound (Eq. (23)) with appropriate parameters to obtain the claimed error guarantee. Recall that we condition on the event $\mathcal{E}$ that all relevant thresholds satisfy $R_{x,j} < B$, which occurs with probability $1 - \delta$ at all times $t \in [T]$.

Then, conditioning on $\mathcal{E}$, observe that with probability at least $1 - \delta$, for all $t \in [T]$ we have

$$|\tilde{F}^{(t)} - F^{(t)}| \le C_1 \cdot B \cdot \frac{\log^{3/2} T}{\varepsilon_{\mathrm{dp}}} \log \frac{T}{\delta} + C \left(\frac{B}{\varepsilon_{\mathrm{dp}}} \log\left(\frac{T}{\delta}\right) + \varepsilon_{\mathrm{dp}} \cdot P^{(t)}\right) + \varepsilon' \cdot F_{\max}$$

We select parameters as follows:

$$\varepsilon' = \Theta(\varepsilon)$$
$$\varepsilon_{\mathrm{dp}} := \Theta(\varepsilon)$$
$$\tau := \frac{\varepsilon^2 F_{\max}}{\log^{7/2}(T) \cdot \log^2(1/\delta)}$$

Recall that $B := \tau \cdot \log\left(\frac{T}{\delta}\right)$. Then, by this choice of parameters (and absorbing the constants into $\varepsilon$), it follows that

$$|\tilde{F}^{(t)} - F^{(t)}| \leq C_1 \cdot \frac{\tau}{\varepsilon} \cdot \log^{7/2}(T) \log^2\left(\frac{1}{\delta}\right) + C\left(\frac{\tau}{\varepsilon}\log^2(T)\log^2\left(\frac{1}{\delta}\right) + \varepsilon F_{\max}\right) + \varepsilon F_{\max}$$

$$\leq \varepsilon \cdot F_{\max}$$

**Space Complexity.** We analyze the total space used by the algorithm. Throughout the stream, the algorithm stores the sample $S_t$ and the associated counts $c_x$ for $x \in S_t$, in addition to $O(\log T)$ tree mechanism counters.

By Lemma D.5, the size of the sample $S_t$ maintained by Algorithm 3 is bounded by

$$|S_t| \leq \frac{F_t}{\tau} + \sqrt{\frac{2F_t}{\tau}\log\frac{1}{\delta}} + \frac{2}{3}\log\frac{1}{\delta} \leq C' \cdot \frac{1}{\varepsilon^2} \cdot \log^{7/2}(T) \cdot \log^2(1/\delta)$$

for some universal constant $C' > 0$. Thus, the algorithm stores $O\left(\frac{1}{\varepsilon^2} \cdot \log^{9/2}(T) \cdot \log^2(1/\delta)\right)$ bits of memory in total for all of the counts of sampled elements. Note that the $O(\log T)$ counters needed for the tree mechanism are dominated by this term, so the total space bound follows.

$\square$

### D.5. Achieving the Prefix-Max Error Guarantee

In this subsection, we modify the robust algorithm from Section D.2 to achieve the prefix-max error guarantee and finish the proof of Theorem 3.1. For notational convenience, let $M_t = \max_{t' \leq t} F_{t'}$. We design a modified estimator $\tilde{F}_t$ which satisfies

$$|\tilde{F}_t - F_t| \leq \varepsilon \cdot M_t \quad \text{for all } t \in [T]$$

Let $\mathcal{A}_1, ..., \mathcal{A}_L$ be independent instances of the robust $\ell_1$ estimation algorithm from Section D.2 for $L = O(\log M_T) = O(\log T)$, where $\mathcal{A}_k$ is instantiated with threshold parameter

$$\tau_k := \frac{\varepsilon^2 \cdot 2^k}{\log^{7/2}(T) \cdot \log^2(1/\delta)}$$

and $\tilde{F}_t^{(k)}$ is the estimate returned by $\mathcal{A}_k$ at time $t \in [T]$.

---

**Algorithm via Sketch-Switching**:

1. Initialize counter $c = 1$ to indicate the current active sketch $\mathcal{A}_c$.

2. For stream update $(x, \Delta)$ at time $t$:

   (a) Update each active copy among $\mathcal{A}_1, \dots, \mathcal{A}_L$ with $(x, \Delta)$.

   (b) We *activate* a copy $\mathcal{A}_k$ at the first time $t$ when $\tilde{F}_t^{(k)} \geq 2^k$. Return $\tilde{F}_t := \tilde{F}_t^{(c)}$, where $c = \max\{k : \mathcal{A}_k \text{ is active at time } t\}$.

   (c) De-allocate space from $\mathcal{A}_k$ for $k < c$ (instances which are no longer active).

---

*Proof of Theorem 3.1.* We first recall the error bound for a single robust $\ell_1$ estimation sketch $\mathcal{A}_k$ from Section D.2, now written in terms of the prefix-max error guarantee. Observe that by Corollary D.7, we know that

$$|\hat{D}_t - D_t| \leq \varepsilon\sqrt{2^k \cdot F_t} + \varepsilon^2 \cdot 2^k$$

By combining this with Eq. (22), we have

$$|\hat{F}_t - F_t| \leq |\hat{P}_t - P_t| + |\hat{D}_t - D_t|$$

$$\leq C \cdot \left(\frac{B}{\varepsilon}\log\left(1 + \frac{T}{\delta}\right) + \varepsilon \cdot P_t\right) + \frac{1}{2}\varepsilon \cdot 2^k + \varepsilon\sqrt{2^k \cdot F_t}$$

$$\leq C \cdot \frac{B}{\varepsilon}\log\left(1 + \frac{T}{\delta}\right) + O(\varepsilon) \cdot (2^k + F_t) + \varepsilon\sqrt{2^k \cdot F_t}$$

where the second inequality follows since $P_t \leq F_t$. Then, by applying the error guarantee from the Tree Mechanism (Eq. (21)), it follows that

$$|\tilde{F}_t - F_t| \leq |F_t - \hat{F}_t| + |\hat{F}_t - \tilde{F}_t|$$

$$\leq C \cdot \frac{B}{\varepsilon}\log\left(1 + \frac{T}{\delta}\right) + O(\varepsilon) \cdot (2^k + F_t) + \varepsilon\sqrt{2^k \cdot F_t} + C_1 \cdot B \cdot \frac{\log^{3/2} T}{\varepsilon}\log\frac{T}{\delta}$$

$$\leq O\left(\tau \cdot \log^2 T \cdot \log^2 \frac{1}{\delta}\right) + O(\varepsilon) \cdot (2^k + F_t) + \varepsilon\sqrt{2^k \cdot F_t} + O\left(\tau \cdot \log^{7/2} T \cdot \log^2 \frac{1}{\delta}\right)$$

For $B = \tau_k \cdot \log(\frac{T}{\delta})$ and $\tau_k$ chosen as above (and absorbing constants into $\varepsilon$), we see that $\mathcal{A}_k$ satisfies

$$|\tilde{F}_t^{(k)} - F_t| \leq \varepsilon \cdot (2^k + F_t) + \varepsilon\sqrt{2^k \cdot F_t}$$

By the sketch-switching construction, we use the output of the largest active instance $\mathcal{A}_c$, i.e. $\tilde{F}_{t'}^{(c)} \geq 2^c$ at some time $t' \leq t$ and for all $k > c$, we have $\tilde{F}_{t'}^{(k)} < 2^k$ at all times $t' \leq t$. Note that if $\tilde{F}_{t'}^{(c)} \geq 2^c$ at some time $t' \leq t$, we have

$$F_{t'} \geq 2^c - \varepsilon(2^c + F_{t'}) - \varepsilon\sqrt{2^c \cdot F_{t'}}$$

By solving the inequality above for $F_{t'}$, we obtain

$$F_{t'} \geq (1 - 4\varepsilon) \cdot 2^c$$

Alternatively, we know that for $k > c$, $\tilde{F}_{t'}^{(k)} < 2^k$ for all $t' \leq t$, it follows that

$$F_{t'} \leq 2^k + \varepsilon(2^k + F_{t'}) + \varepsilon\sqrt{2^k \cdot F_{t'}}$$

By solving the inequality above, we obtain that

$$F_{t'} \leq (1 + 4\varepsilon) \cdot 2^k$$

To summarize, if $F_{t'} \geq (1 - 4\varepsilon) \cdot 2^c$ at any previous time $t' \leq t$ in the stream and $F_{t'} \leq (1 + 4\varepsilon) \cdot 2^k$ for all $t' \leq t$ for $k > c$, we use sketch $\mathcal{A}_c$. Equivalently, we use sketch $\mathcal{A}_c$ at time $t$ if and only if

$$(1 - 4\varepsilon) \cdot 2^c \leq \max_{t' \leq t} F_{t'} \leq (1 + 4\varepsilon) \cdot 2^{c+1}$$

Therefore, we have shown that

$$|\tilde{F}_t - F_t| \leq O(\varepsilon) \cdot M_t$$

as claimed.

**Robustness.** Robustness follows from the robustness of each $\mathcal{A}_k$ to adaptive inputs.

**Space Complexity.** Recall that by Lemma D.5, each copy $\mathcal{A}_k$ stores an evolving sample $S_t^{(k)}$, for threshold parameter $\tau_k$ and failure probability $\gamma$. Substituting our choice of parameters, we have

$$|S_t^{(k)}| \le \frac{F_t}{\tau_k} + \sqrt{\frac{2F_t}{\tau_k} \log \frac{1}{\gamma}} + \frac{2}{3} \log \frac{1}{\gamma}$$

$$\le \frac{\log^{7/2}(T) \log^2\left(\frac{1}{\delta}\right) \cdot F_t}{\varepsilon^2 \cdot 2^k} + \sqrt{\frac{\log^{7/2}(T) \log^3\left(\frac{1}{\delta}\right) \cdot F_t}{\varepsilon^2 \cdot 2^k}} + \frac{2}{3} \log \frac{1}{\delta} \le C \cdot \frac{\log^{7/2} T \log^2\left(\frac{1}{\delta}\right) \cdot F_t}{\varepsilon^2 \cdot 2^k}$$

for some universal constant $C > 0$. Since we stop storing sketch $\mathcal{A}_k$ as soon as $F_t \ge (1 - 4\varepsilon) \cdot 2^c$ for some $c > k$, it follows that the space allocated to storing $\mathcal{A}_k$ is $O\left(\frac{1}{\varepsilon^2} \log^{9/2}(T) \cdot \log^2\left(\frac{1}{\delta}\right)\right)$. The final algorithm stores $L = O(\log T)$ copies of the robust algorithm from Section D.2, so the final space bound follows. $\qquad\square$

# E. Bernstein Statistics

In this section we present our robust restatble sketches for Bernstein (soft concave sublinear) statistics. We first restate the definition:

**Definition 4.1** (Bernstein Functions). We denote by $\mathcal{B}$ (also referred to as the *soft concave sublinear* in (Cohen, 2016; Cohen and Geri, 2019)) the set of all functions $f(w)$ that admit a Lévy-Khintchine representation with zero drift. Formally, $f \in \mathcal{B}$ if there exists a non-negative weight function $a(t) \ge 0$, referred to as the *inverse complement Laplace transform*, such that:

$$f(w) = \mathcal{L}^c[a](w) := \int_0^\infty a(t)\left(1 - e^{-wt}\right) dt.$$

Useful Bernestein functions include:

- *Low Frequency Moments ($w^p, p \in (0,1)$):*

$$a(t) = \frac{p}{\Gamma(1-p)} t^{-(1+p)}.$$

- *Logarithmic Growth ($\ln(1+w)$):*

$$a(t) = \frac{e^{-t}}{t}.$$

- *Soft Capping ($T(1 - e^{-w/T})$):*

$$a(t) = T \cdot \delta(t - 1/T).$$

Additionally, soft concave sublinear statistics provide a constant approximation of concave sublinear statistics, including capping functions.

We prove Theorem 4.2, restated here for convenience:

**Theorem 4.2** (Robust Resettable Sketches for Bernstein Statistics). *Let $f \in \mathcal{B}$ be a Bernstein function. For any accuracy parameter $\varepsilon \in (0,1)$, failure probability $\delta \in (0,1)$, a parameter $T$ that bounds the number of $\mathsf{Inc}$ operations, and a fixed polynomial so that update values are in a range $\Delta \in [\Delta_{\min}, \Delta_{\max}]$ where $\Delta_{\max}/\Delta_{\min} = O(\mathrm{poly}(T))$. There exists a resettable streaming sketch using space*

$$k = O\left(\mathrm{poly}(\varepsilon^{-1}, \log(T/\delta))\right)$$

*that, with probability at least $1 - \delta$ against an adaptive adversary, maintains an estimate $\hat{F}_t$ satisfying for all time steps $t$:*

$$|F_t - \hat{F}_t| \le \varepsilon \max_{t' \le t} F_{t'}.$$

We adapt the framework of Cohen (2016), which reduces composable sketching of Bernstein statistics over unaggregated datasets with nonnegative updates to composable sketching of two base statistics: *Sum*, on the original dataset, and *Max-Distinct* (a generalization of cardinality), on a (randomized) mapped dataset $E$. Any Bernstein statistic $f \in \mathcal{B}$ can be approximated, within a relative error, by a linear combination of these two base statistics. Therefore, sketches with relative error approximation for the base statistics yield a sketch with relative error guarantees for the Bernstein statistic. We review the reduction and its properties in Section E.1.

The adoption of this framework for designing robust and resettable sketches required overcoming some technical challenges:

The weighting of the two base base statistics proposed in (Cohen, 2016) depends on the sum statistic value. We propose (Corollary E.6) and analyze a different parametrization so that the weighting does not depend on the input.

The composable Max-Distinct sketch proposed in (Cohen, 2016) is not robust to input adaptivity (As discussed in Section A.2, composable cardinality sketches are inherently not robust (Cohen et al., 2024; Gribelyuk et al., 2024)). In Section E.3 we propose an approximation of the Bernstein statistic as a linear combination of sum and cardinality statistics. We propose a different map of the input data elements into"output" sets of data elements ($E_i$) so that the component that was approximated by the Max-Distinct statistic is instead approximated by a linear combination of cardinality statistics on datasets ($E_i$). Our map uses independent randomness per generated data element, which facilitates the robustness guarantees.

We conclude the proof of Theorem 4.2 in Section E.4. The modified parametrization and map allow us to approximate the Bernstein statistics (over increments-only datasets) as a linear combination of sum and cardinality statistics (over mapped output datasets). In order to apply this with resettable streams, we show how to implement reset operations on the input streams as reset operations on the output streams so that the approximation guarantee holds. We then specify a Bernstein sketch as a sum sketch applied to the input stream and cardinality sketches applied to the output streams ($E_i$), with the estimator being a respective linear combination of the sketch-based approximations of the base sketches. We establish end-to-end robustness guarantees and set parameters for the base sketches so that prefix-max guarantees of the sum and cardinality sketches transfer to the Bernstein sketch.

### E.1. Bernstein Statistics to Sum and Max-Distinct

We review the reduction of (Cohen, 2016). Let the input dataset $W$ be a set of elements (updates) $e = (x, \Delta)$, where $x$ is a key from a universe $\mathcal{U}$ and $\Delta \geq 0$ is a positive increment value. We define the aggregated weight $w_x$ of a key $x$ as the sum of increments for that key:

$$w_x = \sum_{(x,\Delta)\in W} \Delta.$$

We are interested in estimating statistics of the form $F(W) = \sum_x f(w_x)$, where $f \in \mathcal{B}$.

**Definition E.1** (Complement Laplace Transform)**.** For an aggregated dataset $W = \{(x, w_x)\}$, the complement Laplace transform at parameter $t > 0$ is defined as:

$$\mathcal{L}^c[W](t) := \sum_x \left(1 - e^{-w_x t}\right).$$

In the continuous view, where $W(w)$ denotes the density of keys with weight $w$, this is equivalent to:

$$\mathcal{L}^c[W](t) \equiv \int_0^\infty W(w)\left(1 - e^{-wt}\right)\,dw = \mathsf{Distinct}(W) - \mathcal{L}[W](t), \tag{24}$$

where $\mathcal{L}[W](t)$ is the standard Laplace transform of the frequency distribution.

For a function $f \in \mathcal{B}$ with inverse transform $a(\cdot)$ (see Definition 4.1), we can express the statistic as:

$$F(W) = \int_0^\infty a(t)\mathcal{L}^c[W](t)\,dt. \tag{25}$$

The method of (Cohen, 2016) approximates (25) using sum and Max-Distinct statistics. The Max-Distinct statistic generalizes the Distinct Count (cardinality) statistic:

**Definition E.2** (The Max-Distinct Statistic). Let $E$ be a dataset of pairs $(z, \nu)$, where $z$ is a key and $\nu \geq 0$ is a value. The *Max-Distinct* statistic is defined as the sum over distinct keys of the maximum value associated with each key:

$$\mathsf{MxDist}(E) \ := \ \sum_{z \in \mathsf{Distinct}(E)} \max_{(z,\nu) \in E} \nu.$$

The $\mathsf{Sum}$ statistic is over the original dataset $W$. The $\mathsf{MxDist}$ statistics is of a new stream $E = \bigcup_{e \in W} \mathcal{M}(e)$, obtained by a randomized mapping of each input element $e = (x, \Delta) \in W$ to a set of output elements. The map is described in Algorithm 4, parameterized by a granularity $r \geq 1$ and a cutoff $\tau \geq 0$.

---

**Algorithm 4:** Element Mapping $\mathcal{M}(x, \Delta)$

---

**Input:** Update $(x, \Delta) \in W$, transform $a(t)$, parameters $r, \tau$
**Output:** Output elements emitted to dataset $E$
Draw $r$ independent variables $y_1, \ldots, y_r \sim \mathrm{Exp}(\Delta)$;
**foreach** $k \in \{1, \ldots, r\}$ **do**
    Compute weight $\nu_k \leftarrow \int_{\max(\tau, y_k)}^{\infty} a(t)\, dt$;
    **if** $\nu_k > 0$ **then**
        Generate output key $z_k \leftarrow H(x, k)$ ;        `// Unique key for (input key, index) pair`
        Emit output element $(z_k, \nu_k)$ to dataset $E$ ;

---

The estimator, for a parameter $\tau > 0$, is defined as:

$$\tilde{F}(W) \ = \ \underbrace{\mathsf{Sum}(W) \cdot \int_0^\tau a(t)t\, dt}_{\text{Linear approximation for } t \in [0, \tau]} + \underbrace{\frac{1}{r} \mathsf{MxDist}(E)}_{\text{Sampled tail for } t \in [\tau, \infty)} . \tag{26}$$

The first term approximates the integral (25) in the range $[0, \tau]$, while the second term (a random variable) estimates the range $[\tau, \infty)$ using the stream $E$. Intuitively, Eq. (24) behaves like a linear function (approximated by sum) for low $t$ and like a distinct count for high $t$.

CONCENTRATION ANALYSIS

We bound the error of $\tilde{F}(W)$. The estimator splits the integration domain at $\tau$. We analyze the error contribution of the two ranges separately.

**Claim E.3** (Low-range Bias). Let $\varepsilon, \beta \in (0, 1)$. For any $\tau \leq \frac{\sqrt{\varepsilon}}{\mathsf{Max}(W)}$, the linear approximation error is at most $\varepsilon$. with probability at least $1 - \beta$, the first term of the estimator $\tilde{F}(W)$ approximates $\int_0^\tau a(t)\mathcal{L}^c[W](t)\, dt$ within relative error $\varepsilon$.

*Proof.* The first term $\mathsf{Sum}(W) \int_0^\tau a(t)t\, dt$ relies on the approximation $1 - e^{-w_x t} \approx w_x t$. The relative error of this approximation is bounded by $w_x t$ for small inputs. To ensure the error is at most $\varepsilon$ for *all* keys in the stream simultaneously, we require:
$$w_x t \ \leq \ \sqrt{\varepsilon} \quad \forall x \in W, \forall t \in [0, \tau].$$

This condition holds if and only if the maximum integration limit satisfies $\tau \leq \frac{\sqrt{\varepsilon}}{\max_x w_x}$. Since our chosen $\tau$ satisfies $\tau \leq \frac{\sqrt{\varepsilon}}{\mathsf{Max}(W)}$, the linear approximation is valid, and the integral over $[0, \tau]$ has relative error at most $\varepsilon$ (Lemma 2.1 in (Cohen, 2016)). $\qquad\square$

**Claim E.4** (High-range Variance). Let $\varepsilon, \beta \in (0, 1)$ and for $Y \geq \mathsf{Sum}(W)$ let the granularity $r \geq \Theta(\varepsilon^{-2.5} \log(1/\beta) Y / \mathsf{Sum}(W))$. Then for $\tau \geq \frac{\sqrt{\varepsilon}}{Y}$, the sampling failure probability is at most $\beta$.

*Proof.* The second term estimates the tail integral via sampling. By Lemma 5.1 of (Cohen, 2016), the failure probability is bounded by
$$2 \exp(-r\varepsilon^2 \mathcal{L}^c[W](\tau)/3). \tag{27}$$

We need to ensure the exponent is large enough, i.e., $\mathcal{L}^c[W](\tau) \gtrsim \sqrt{\varepsilon}$. Since $\mathcal{L}^c[W](t)$ is a monotonically increasing function of $t$, it suffices to bound it at the minimum allowed cutoff $\tau_{\min} = \frac{\sqrt{\varepsilon}}{Y}$.

Using Lemma 3.3 of (Cohen, 2016):

$$\mathcal{L}^c[W](\tau) \ \geq \ \left(1 - \frac{1}{e}\right) \mathsf{Sum}(W) \min\left\{\frac{1}{\mathsf{Max}(W)}, \tau\right\}.$$

Substituting $\tau = \tau_{\min} = \frac{\sqrt{\varepsilon}}{Y}$:

$$\mathcal{L}^c[W](\tau_{\min}) \ \geq \ \left(1 - \frac{1}{e}\right) \mathsf{Sum}(W) \min\left\{\frac{1}{\mathsf{Max}(W)}, \frac{\sqrt{\varepsilon}}{Y}\right\}$$
$$= \ \left(1 - \frac{1}{e}\right) \min\left\{\frac{\mathsf{Sum}(W)}{\mathsf{Max}(W)}, \sqrt{\varepsilon}\frac{\mathsf{Sum}(W)}{Y}\right\}.$$

Since $\mathsf{Sum}(W) \geq \mathsf{Max}(W)$ and $\varepsilon < 1$ and $Y \geq \mathsf{Sum}(W)$, the minimum is $\sqrt{\varepsilon}\mathsf{Sum}(W)/Y$. Thus:

$$\mathcal{L}^c[W](\tau) \ \geq \ \mathcal{L}^c[W](\tau_{\min}) \ \geq \ \left(1 - \frac{1}{e}\right)\sqrt{\varepsilon}\frac{\mathsf{Sum}(W)}{Y}.$$

With $r = \Theta(\varepsilon^{-2.5}\log(1/\beta)Y/\mathsf{Sum}(W))$, the concentration exponent Eq. (27) becomes $O(\log(1/\beta))$, ensuring the failure probability is at most $\beta$. $\qquad\square$

**Corollary E.5** (Concentration of the Reduction (Cohen, 2016)). *Let $\varepsilon, \beta \in (0,1)$ and granularity $r = \Theta(\varepsilon^{-2.5}\log(1/\beta))$. For any cutoff $\tau$ in the range:*

$$\tau \ \in \ \left[\frac{\sqrt{\varepsilon}}{\mathsf{Sum}(W)}, \ \frac{\sqrt{\varepsilon}}{\mathsf{Max}(W)}\right],$$

*with probability at least $1 - \beta$, the estimator $\tilde{F}(W)$ approximates $F(W)$ within relative error $\varepsilon$.*

*Proof.* We combine Claim E.3 and Claim E.4, setting $Y = \mathsf{Sum}(W)$. Since any $\tau$ in the specified interval satisfies both the upper bound required for the bias (Claim E.3) and the lower bound required for the variance (Claim E.4), the total error is bounded by $\varepsilon F(W)$ with probability $1 - \beta$. $\qquad\square$

### E.2. Input-independent Parametrization

Consider a dataset $W$ that contains at most $T$ updates with values $\in [\Delta_{\min}, \Delta_{\max}]$.

**Corollary E.6** (Input-independent threshold). *Let $\varepsilon, \beta \in (0,1)$. For cutoff $\tau = \frac{\sqrt{\varepsilon}}{T\Delta_{\max}}$ and granularity $r = \Theta(\varepsilon^{-2.5}\log(1/\beta)T\Delta_{\max}/\Delta_{\min})$, with probability at least $1 - \beta$, the estimator $\tilde{F}(W)$ (26) approximates the Bernstein statistics $F(W)$ within relative error $\varepsilon$.*

*Proof.* For the low range accuracy, apply Claim E.3 observing that since $T\Delta_{\max} \geq \mathsf{Sum}(W) \geq \mathsf{Max}(W)$ and hence $\tau \leq \sqrt{\varepsilon}/\mathsf{Max}(W)$.

For the high range accuracy, apply Claim E.4 with $Y := T\Delta_{\max}$. Observe that $Y \geq \mathsf{Sum}(W)$ and that $\mathsf{Sum}(W) > 0 \implies Y/\Delta_{\min} \geq Y/\mathsf{Sum}(W)$ on a dataset with at most $T$ updates. $\qquad\square$

### E.3. Max-Distinct to Cardinality

We now introduce a new approximation of the tail part

$$I := \int_{\tau}^{\infty} a(t)\mathcal{L}^c[W](t)\,dt \tag{28}$$

that has the form of a linear combination of cardinality statistics:

$$\tilde{I} := \frac{1}{r}\sum_{i \in [m]} \alpha_i \mathsf{Distinct}(E_i). \tag{29}$$

---

**Algorithm 5:** Stream Element Mapping $\mathcal{M}^*(x, \Delta)$

---

**Input:** Update $(x, \Delta) \in W$, parameters $r$, $\tau = \tau_0 < \cdots < \tau_m$
**Output:** Output elements emitted to Streams $E_1, \ldots, E_m$
**for** $k \in \{1, \ldots, r\}$ **do**
 Generate output key $z_k \leftarrow H(x, k)$ ;        // Unique key for (input key, index) pair
 **for** $i \in \{1, \ldots, m\}$ **do**
  Sample independent $y \sim \text{Exp}(\Delta)$;
  **if** $y < \tau_i$ **then**
   Emit output element $\mathsf{Insert}(z_k)$ to Stream $E_i$

---

The datasets $(E_i)_{i \in [m]}$ are generated by the randomized map Algorithm 5 that is parametrized by $r$ and $(\tau_i)_{i=0}^m$. The thresholds $(\tau_i)_{i=0}^m$ and the weights $\alpha_i$ are determined as follows from $\tau$.

Let $V(t) = \int_t^\infty a(x) \, dx$ be the value function. Define the sequence of target values $v_0, v_1, \ldots, v_m$ geometrically:

$$v_i = (1 + \varepsilon)^{-i} V(\tau).$$

Set the thresholds $\tau_i$ to track these values:

$$\tau_i = \sup\{t : V(t) \geq v_i\}.$$

(If the set is empty, $\tau_i = \infty$). Truncate the sequence at $m$ such that $v_m \leq \frac{\varepsilon}{T} f(\Delta_{\min})$.

Define weights $\alpha_i := V(\tau_{i-1}) - V(\tau_i)$ for $i \in [m]$.

### E.3.1. BOUND THE NUMBER OF TERMS

We bound the number of terms in Eq. (29). We use the following:

**Claim E.7** (Tail Integral Bound). *For any Bernstein function $f \in \mathcal{B}$ and cutoff $\tau = \frac{\sqrt{\varepsilon}}{T\Delta_{\max}}$, the tail integral of the measure is bounded by:*

$$\int_\tau^\infty a(t) \, dt \leq \frac{2}{\sqrt{\varepsilon}} \cdot \frac{T\Delta_{\max}}{\Delta_{\min}} \cdot f(\Delta_{\min}). \tag{30}$$

*Proof.* First, we relate the tail integral to the function value $f(1/\tau)$. Recall that $f(x) = \int_0^\infty (1 - e^{-xt}) a(t) \, dt$. Restricting the integral to $t \geq \tau$ and setting $x = 1/\tau$, we have:

$$f(1/\tau) \geq \int_\tau^\infty (1 - e^{-t/\tau}) a(t) \, dt.$$

For $t \geq \tau$, we have $t/\tau \geq 1$, which implies $1 - e^{-t/\tau} \geq 1 - e^{-1}$. Thus:

$$f(1/\tau) \geq (1 - e^{-1}) \int_\tau^\infty a(t) \, dt \implies \int_\tau^\infty a(t) \, dt \leq \frac{1}{1 - e^{-1}} f(1/\tau).$$

Next, we use the subadditivity of Bernstein functions ($f(\lambda x) \leq \lambda f(x)$ for $\lambda \geq 1$) to scale from $\Delta_{\min}$ to $1/\tau$. Note that since $\Delta_{\max} \geq \Delta_{\min}$ and $\varepsilon \in (0, 1)$, the factor $\lambda = \frac{1}{\tau \Delta_{\min}}$ is $\geq 1$. Thus:

$$f(1/\tau) = f\left(\frac{1}{\tau \Delta_{\min}} \cdot \Delta_{\min}\right) \leq \frac{1}{\tau \Delta_{\min}} f(\Delta_{\min}).$$

Substituting $\tau = \frac{\sqrt{\varepsilon}}{T\Delta_{\max}}$ into the expression and noting that $(1 - e^{-1})^{-1} \approx 1.58 \leq 2$ yields the claim. $\square$

**Lemma E.8** (Bound on the number of terms). *For any Bernstein function $f \in \mathcal{B}$, dataset size $T$, and cutoff $\tau = \frac{\sqrt{\varepsilon}}{T\Delta_{\max}}$, $m = O(\varepsilon^{-1} \log T)$.*

*Proof.* If $m \leq 2$, the bound holds trivially. Assume $m > 2$. Since the cutoff was not at step $m - 1$, it holds that:

$$v_{m-1} > \frac{\varepsilon}{T} f(\Delta_{\min}).$$

The number of levels $m$ is determined by the ratio of the start value $v_0$ to $v_{m-1}$:

$$(1 + \varepsilon)^{m-1} = \frac{v_0}{v_{m-1}} < \frac{v_0}{\frac{\varepsilon}{T} f(\Delta_{\min})}.$$

Substituting the bound for $v_0$ from Claim E.7 ($v_0 \leq O(\varepsilon^{-0.5}) \frac{T \Delta_{\max}}{\Delta_{\min}} f(\Delta_{\min})$):

$$(1 + \varepsilon)^{m-1} < O\left(\varepsilon^{-1.5}\right) \cdot T^2 \cdot \frac{\Delta_{\max}}{\Delta_{\min}}.$$

Taking logarithms and using $\Delta_{\max}/\Delta_{\min} = \text{poly}(T)$ (as assumed in the statement on Theorem 4.2), we obtain $m = O(\varepsilon^{-1} \log T)$. $\qquad\square$

The number of terms $m$ can be lower for some functions (which yields an improved storage bound):

- **Moments ($w^p$):** The values scale polynomially, requiring $m = O(\varepsilon^{-1} \log T)$.

- **Logarithmic ($\ln(1 + w)$):** The values scale logarithmically, requiring $m = O(\varepsilon^{-1} \log \log T)$.

- **Soft Capping:** The value is constant (step function), requiring $m = 1$.

### E.3.2. ACCURACY

**Lemma E.9** (Approximation via Cardinality). *Let the granularity $r$ and $\tau$ be set as in Corollary E.6. Let $\tau_0, \ldots, \tau_m$ be the thresholds defined by the value levels $v_i = (1 + \varepsilon)^{-i} V(\tau_0)$ and let $\alpha_i = v_{i-1} - v_i$. Let $\tilde{I}$ as in Eq. (29) be the estimator constructed from the output streams of Algorithm 5 for the tail $I$ as in Eq. (28). With probability at least $1 - \beta$:*

$$\left| \tilde{I} - I \right| \leq 2\varepsilon I + \varepsilon f(\Delta_{\min}).$$

*Proof.* The error consists of the **discretization bias** $|\mathbb{E}[\tilde{I}] - I|$ (due to using a sum instead of an integral) and the **concentration** of $\tilde{I}$ around its expectation $\mathbb{E}[\tilde{I}]$.

**1. Expectation (Discretization bias).** We first analyze the expected value of the estimator. By linearity of expectation:

$$\mathbb{E}[\tilde{I}] = \frac{1}{r} \sum_{i=1}^{m} \alpha_i \mathbb{E}[\text{Distinct}(E_i)].$$

For a specific level $i$, a key $z_k$ corresponding to input $x$ is included in $E_i$ if *any* of the independent updates for $x$ trigger an insertion. The probability of inclusion is $1 - e^{-w_x \tau_i}$. Summing over all $r$ copies and all keys $x$:

$$\mathbb{E}[\text{Distinct}(E_i)] = r \sum_x (1 - e^{-w_x \tau_i}) = r \mathcal{L}^c[W](\tau_i).$$

Substituting this back, $\mathbb{E}[\tilde{I}] = \sum_{i=1}^{m} \alpha_i \mathcal{L}^c[W](\tau_i)$. This sum approximates the integral $I = \int_{\tau_0}^{\infty} a(t) \mathcal{L}^c[W](t) \, dt$ via a Riemann sum.

- **Tail Truncation:** The integral from $\tau_m$ to $\infty$ is omitted. This contributes an additive error of at most $T \cdot v_m \leq \varepsilon f(\Delta_{\min})$.

- **Discretization:** For $t \in (\tau_{i-1}, \tau_i]$, we approximate $\mathcal{L}^c[W](t)$ by the constant $\mathcal{L}^c[W](\tau_i)$. Since the value $V(t)$ drops by at most a factor of $(1 + \varepsilon)$ in this interval, standard Riemann sum bounds yield a relative error of $\varepsilon$.

Thus: $\left| \mathbb{E}[\tilde{I}] - I \right| \leq \varepsilon I + \varepsilon f(\Delta_{\min})$.

**2. Concentration** Let $Z_{x,k,i}$ be the indicator that the unique key associated with input $x$ and repetition $k$ (denoted $z_{x,k}$) is inserted into stream $E_i$. The insertion condition is $\min\{y \sim \text{Exp}(\Delta) \text{ for updates of } x\} < \tau_i$. The probability of this event is $p_{x,i} = 1 - e^{-w_x \tau_i}$. Crucially, since Algorithm 5 uses independent randomness for each level $i$ and each repetition $k$, the variables $Z_{x,k,i}$ are mutually independent for all $x \in \mathcal{U}, k \in [r], i \in [m]$. We can write the estimator as:

$$\tilde{I} = \sum_{i=1}^{m} \sum_{k=1}^{r} \sum_{x \in \mathcal{U}} \frac{\alpha_i}{r} Z_{x,k,i}.$$

Let $Y_{x,k,i} = \frac{\alpha_i}{r} Z_{x,k,i}$. The estimator is $\hat{I} = \sum Y_{x,k,i}$. These are independent bounded random variables with $Y_{x,k,i} \in [0, \frac{\alpha_i}{r}]$. We apply Bernstein's inequality. We need to bound the maximum magnitude $M$ and the total variance $\sigma^2$.

- **Magnitude $M$:** The weights are defined as $\alpha_i = v_{i-1} - v_i$ where $v_i = (1+\varepsilon)^{-i} v_0$. Thus $\alpha_i \leq \varepsilon v_{i-1} \leq \varepsilon v_0$. From Claim E.7, $v_0 = \int_{\tau_0}^{\infty} a(t)dt \leq \frac{2}{\sqrt{\varepsilon}} \frac{T\Delta_{\max}}{\Delta_{\min}} f(\Delta_{\min})$. Thus:

$$M = \max_i \frac{\alpha_i}{r} \leq \frac{\varepsilon v_0}{r}.$$

- **Variance $\sigma^2$:** $\text{Var}(Z_{x,k,i}) = p_{x,i}(1 - p_{x,i}) \leq p_{x,i}$. Summing the variances:

$$\sigma^2 = \sum_{i,k,x} \left(\frac{\alpha_i}{r}\right)^2 \text{Var}(Z_{x,k,i}) \leq \frac{1}{r} \sum_{i=1}^{m} \alpha_i^2 \underbrace{\sum_x p_{x,i}}_{\mathcal{L}^c[W](\tau_i)} = \frac{1}{r} \sum_{i=1}^{m} \alpha_i^2 \mathcal{L}^c[W](\tau_i).$$

Since $\alpha_i \leq \varepsilon v_0$, we can bound the sum:

$$\sigma^2 \leq \frac{\varepsilon v_0}{r} \sum_{i=1}^{m} \alpha_i \mathcal{L}^c[W](\tau_i) = \frac{\varepsilon v_0}{r} \mathbb{E}[\tilde{I}].$$

**Applying the Bound.** We seek to bound the probability that $|\tilde{I} - \mathbb{E}[\tilde{I}]| > \varepsilon \max\{\mathbb{E}[\tilde{I}], f(\Delta_{\min})\}$. Bernstein's inequality states:

$$\Pr[|\tilde{I} - \mathbb{E}[\tilde{I}]| > \lambda] \leq 2\exp\left(-\frac{\lambda^2}{2\sigma^2 + 2M\lambda/3}\right).$$

We first consider the case $\mathbb{E}[\tilde{I}] \geq f(\Delta_{\min})$. Set $\lambda = \varepsilon \mathbb{E}[\tilde{I}]$. We obtain that the exponent is $\Theta\left(\frac{r\varepsilon \mathbb{E}}{v_0}\right)$

Using $\mathbb{E} \geq f(\Delta_{\min})$ and the bound for $v_0$:

$$\frac{r\varepsilon \mathbb{E}[\tilde{I}]}{v_0} \geq r\varepsilon \frac{f(\Delta_{\min})}{\frac{2}{\sqrt{\varepsilon}} \frac{T\Delta_{\max}}{\Delta_{\min}} f(\Delta_{\min})} = r\varepsilon^{1.5} \frac{\Delta_{\min}}{2T\Delta_{\max}}.$$

To ensure the probability is $\leq \beta$, we require this term to be $\Omega(\log(1/\beta))$. This is satisfied if:

$$r \geq \Theta\left(\frac{1}{\varepsilon^{1.5}} \frac{T\Delta_{\max}}{\Delta_{\min}} \log(1/\beta)\right).$$

The granularity $r$ specified in Corollary E.6 suffices to satisfy this.

The calculation for the case $\mathbb{E}[\tilde{I}] \leq f(\Delta_{\min})$ is similar: We set $\lambda = \varepsilon f(\Delta_{\min})$. The exponent is $\Omega\left(\frac{r f(\Delta_{\min})}{v_0}\right)$. Substituting the values for $v_0$ and then $r = \Omega\left(\left(\frac{1}{\varepsilon^{0.5}} \frac{T\Delta_{\max}}{\Delta_{\min}} \log(1/\beta)\right)\right)$ we obtain that the exponent is $\Omega(r\varepsilon^{0.5} \frac{\Delta_{\min}}{2T\Delta_{\max}}) = \Omega(1/\beta)$. $\qquad \square$

## E.4. Wrapping it up

We summarize our modified reduction and the properties we use here. We approximate the Bernstein statistic on dataset $W$ by

$$\tilde{F}(W) \;=\; \alpha_0 \cdot \mathsf{Sum}(W) \;+\; \frac{1}{r} \sum_{i \in [m]} \alpha_i \mathsf{Distinct}(E_i). \tag{31}$$

Where $\alpha_i \geq 0$, $\alpha_0 = \int_0^\tau a(t) t \, dt$ and the parameters $(\alpha_i)$ only dependent on $f, \tau$ and the desired accuracy and confidence. The datasets $(E_i)$ are generated by a randomized mapping of each input element to output elements. The generation is independent (a Bernoulli flip that depends on the increment value $\Delta$ of the input element and $i$) for each output element. The number of terms is $m + 1 = O(\varepsilon^{-1} \log T)$.

From Corollary E.6 and Lemma E.9 (by adjusting constants), with probability at least $1 - \beta$, the approximation is within relative error $\varepsilon$ and additive error $\varepsilon f(\Delta_{\min})$. Since when $F(W) > 0$, $F(W) > f(\Delta_{\min})$ we obtain that, a lightly modified estimator (if the estimate is $\tilde{F}(W) < (1 - \varepsilon) f(\Delta_{\min})$, replace it with 0 and otherwise report it) has a relative error of $\varepsilon$.

We observe that since the map Algorithm 5 is applied independently to each input data element, it applies when the input data elements $e \in W$ are streamed and the respective output elements are emitted to the output streams $(E_i)$. The approximation guarantee then holds at each time step.

What remains to incorporate reset operations, and specify a Bernstein sketch that uses as ingredients robust resettable sketches for the base streams so that robustness and prefix-max guarantees transfer.

### E.4.1. INCORPORATING RESETS

The map and estimator Eq. (31) were specified for an incremental input stream and generated output streams with $\mathsf{Insert}$ operations. We implement a $\mathsf{Reset}(x)$ operation in the input stream by emitting $\mathsf{Delete}(H(x,k))$ for $k \in [r]$ for each of the output stream $(E_i)$. The resets perfectly undo the contribution to the distinct statistics of the mapped output elements generated by all prior $\mathsf{Inc}(\mathsf{x}, \Delta$ input elements. Therefore, the relative error approximation guarantee established for Eq. (31) still holds with a resettable input stream.

### E.4.2. THE BERNSTEIN SKETCH

We design a resettable Bernstein sketch based on Eq. (31). We apply a resettable sketch for sum (on the input stream) and $m$ cardinality sketches (for each of the output streams). We then publish a linear combination of the running estimates

$$\hat{F}(W) \;=\; \alpha_0 \cdot \widehat{\mathsf{Sum}}(W) \;+\; \frac{1}{r} \sum_{i \in [m]} \alpha_i \widehat{\mathsf{Distinct}}(E_i).$$

where $\widehat{\mathsf{Sum}}$ and $\widehat{\mathsf{Distinct}}$ are the respective published estimates of sketches that process the resettable streams $W$ and $(E_i)_{i \in [m]}$.

For sum, we use our adjustable-threshold resettable sum sketch in Section D. We use parameters $\varepsilon' = \varepsilon/(m+1)$, $\delta' = \delta/(m+1)$, and input stream with at most $T$ $\mathsf{Inc}(\cdot, \cdot)$ updates. The sketch size is $O(\varepsilon'^{-2} \mathrm{polylog}(T/\delta')) = O(\varepsilon^{-4} \mathrm{polylog}(T/\delta))$.

For cardinality, we use our adjustable rate robust resettable sketch as in Theorem 2.4 (Algorithm 2) with parameters $\varepsilon' = \varepsilon/(m+1)$, $T' = Tr$, and $\delta' = \delta/(m+1)$. Using that $r = O(\mathrm{poly}(T))$ we obtain a space bound of $O(\varepsilon'^{-2} \mathrm{polylog}(T/\delta')) = O(\varepsilon^{-4} \mathrm{polylog}(T/\delta))$ for each sketch.

Since there are $m + 1 = O(\varepsilon^{-1} \mathrm{polylog}(T))$ sketches, the overall space bound of the Bernstein sketch is $O(\mathrm{poly}(\varepsilon^{-1}, \log(T/\delta)))$.

We now analyze the approximation error. At a step where the prefix-max of the Bernstein statistic is $F_{\max}$, the prefix-max value of the sum statistic is bounded by $F_{\max}/\alpha_0$. Per the prefix-max approximation guarantee, the sum sketch returns estimates that are within an additive error of $\varepsilon' F_{\max}/\alpha_0$. These estimates contribute to the error of the linear combination estimator at most $\alpha_0 \cdot \varepsilon' F_{\max}/\alpha_0 = \varepsilon' F_{\max} \leq \varepsilon F_{\max}/(m+1)$.

Similarly, consider the cardinality sketch that is applied to $E_i$. When the Bernstein prefix-max is $F_{\max}$, the cardinality prefix-max value is bounded by $r F_{\max}/\alpha_i$. Per the prefix-max guarantee, the additive error of the estimates is at most $\varepsilon' r F_{max}/\alpha_i$. The contribution to the error of the linear combination estimate is at most $\frac{\alpha_i}{r} \varepsilon' r F_{\max}/\alpha_i = \varepsilon' F_{\max} \leq \varepsilon F_{\max}/(m+1)$.

Combining the additive errors from all the $m + 1$ terms we obtain a total prefix-max error for the Bernstein sketch of $\varepsilon F_{\max}$ that holds with probability $1 - (m + 1)\delta' = 1 - \delta$.

We note that Bernstein statistics that can be approximated with a smaller number of terms $m$ allow for improved storage bounds.

### E.4.3. TRANSFER OF ROBUSTNESS

We establish that the robustness of the sketches for the base statistics implies robustness of the Bernstein sketch. Observe that the only randomness in the reduction Eq. (31) is in the independent random variables in Algorithm 5 that determine whether an insertion to each output stream occurs. In our robustness analysis, we "push" the randomness that is introduced in the map Algorithm 5 into the streams processed by the cardinality sketches so that the robustness analysis of each cardinality sketch applies end to end to the (independent) generation process of each stream $(E_i)$ and the sketch-based reporting of cardinality estimates. We observe that the robust sampling analysis in Lemma C.2 goes through the same way (with the adjusted expectation) when we instead consider each privacy unit to be the Bernoulli that is the combined randomness of (the known to the adversary) $Z_{x,k,i}$, which determines whether an element is emitted into the stream, and the Bernoulli sampler with $p_t$ of the cardinality sketch.

## F. Appendix: DP generalization bound

**Theorem F.1** (DP generalization via stability (Dwork et al., 2015a; Bassily et al., 2021; Feldman and Steinke, 2018))**.** *Let $\mathcal{A}$ be an $\xi$-differentially private algorithm that, given a dataset $S = (x_1, \ldots, x_d) \in X^d$, outputs functions $h_1, \ldots, h_d : X \to [0, 1]$. For any $Y = (y_1, \ldots, y_d) \in X^d$ define*

$$h(Y) := \sum_{i=1}^{d} h_i(y_i).$$

*Let $\mathcal{D}$ be any distribution over $X$, and let $S \sim \mathcal{D}^d$ be an i.i.d. sample. Then for every $\beta \in (0, 1]$,*

$$\Pr_{S \sim \mathcal{D}^d, \, h \leftarrow \mathcal{A}(S)} \left[ e^{-2\xi} \mathop{\mathbb{E}}_{Z \sim \mathcal{D}^d}[h(Z)] - h(S) > \tfrac{4}{\xi} \log(1 + 1/\beta) \right] < \beta,$$

$$\Pr_{S \sim \mathcal{D}^d, \, h \leftarrow \mathcal{A}(S)} \left[ h(S) - e^{2\xi} \mathop{\mathbb{E}}_{Z \sim \mathcal{D}^d}[h(Z)] > \tfrac{4}{\xi} \log(1 + 1/\beta) \right] < \beta.$$

## G. Tree Mechanism

We describe the standard binary tree mechanism (Chan et al., 2011; Dwork et al., 2010; Dwork and Roth, 2014) for privately releasing prefix sums under continual observation. The mechanism is instantiated with a time horizon $T$, a privacy parameter $\varepsilon$, and a sensitivity parameter $L > 0$.

The mechanism processes a stream of updates $(u_t)_{t=1}^{T}$ for $u_t \in \mathbb{R}$, and at each time $t$, outputs a noisy prefix sum $\tilde{S}_t$ which approximates

$$S_t := \sum_{t'=1}^{t} u_{t'}.$$

The noise mechanism is as follows. Let the leaves of a complete binary tree correspond to time steps $1, \ldots, T$, and let each internal node $v$ represent an interval $I_v \subseteq \{1, \ldots, T\}$. For each node $v$ we maintain

$$Z_v = \sum_{t \in I_v} u_t + \eta_v,$$

where the noises $\eta_v$ are independent random variables with $\eta_v \sim \mathrm{Lap}(\lambda)$ and

$$\lambda = \frac{L \log_2 T}{\varepsilon}.$$

At time $t$, we represent the prefix interval $[1..t]$ as a disjoint union of $O(\log T)$ node intervals $\{I_{v_1}, \ldots, I_{v_\ell}\}$ and output

$$\tilde{S}_t := \sum_{j=1}^{\ell} Z_{v_j}.$$

### G.1. Properties of the tree mechanism

**Lemma G.1** (Accuracy of the Tree Mechanism (Chan et al., 2011; Dwork et al., 2010; Dwork and Roth, 2014)). *Let $(\tilde{S}_t)_{t=1}^T$ be the output of the binary tree mechanism with noise scale $\lambda = \frac{L \log T}{\varepsilon}$. There is an absolute constant $C_1 > 0$ such that For any $\delta \in (0, 1)$, with probability at least $1 - \delta$, we have simultaneously for all $t \in \{1, \ldots, T\}$:*

$$|\tilde{S}_t - S_t| \;\leq\; C_1 \frac{L}{\varepsilon} \cdot \log^{3/2} T \cdot \log \frac{T}{\delta}.$$

**Observation G.2** (Streaming storage (Chan et al., 2011; Dwork et al., 2010; Dwork and Roth, 2014)). The tree mechanism can be implemented in a streaming setting with working storage of only $O(\log t)$ active counters at any time $t$. In particular, the maximum working storage is (deterministically) $O(\log T)$.

### G.2. Privacy properties

The standard privacy analysis of the tree mechanism is stated with respect to adjacency on the updates. When each update satisfies $u_t \in [-L, L]$, we have the following:

**Lemma G.3** (Event-level differential privacy (Chan et al., 2011; Dwork et al., 2010; Dwork and Roth, 2014)). *The tree mechanism $(\tilde{S}_t)_{t=1}^T$ is $\varepsilon$-differentially private with respect to event-level adjacency over the stream $(u_t)_{t=1}^T$.*

*That is, for any two update streams $u = (u_1, \ldots, u_T)$ and $u' = (u'_1, \ldots, u'_T)$ that differ in at most one time step $t \in [T]$ and for all sets of outputs $\mathcal{O}$,*

$$\Pr[\mathcal{M}(u) \in \mathcal{O}] \;\leq\; e^\varepsilon \cdot \Pr[\mathcal{M}(u') \in \mathcal{O}].$$

Note that this analysis holds even under adaptivity, where future updates may depend on prior releases of the mechanism.

For our applications, however, the sensitive units are not individual updates: each unit may contribute to multiple time steps, and we explicitly allow portions of updates to be non-sensitive. We assume that each sensitive unit has bounded total contribution in $\ell_1$-norm, possibly chosen adaptively as a function of prior releases.

**Lemma G.4** (Tree mechanism with multiple updates per sensitive unit). *Consider a stream $(u_t)_{t=1}^T$ decomposed as $u_t = \sum_i u_t^{(i)}$, where for each unit $i$ we have*

$$\sum_{t=1}^T |u_t^{(i)}| \;\leq\; L$$

*for some bound $L > 0$. The contributions $u_t^{(i)}$ at time $t$ may depend on all past outputs of the mechanism. Define unit-level adjacency by allowing the entire contribution of a single unit $i$ (i.e., the sequence $(u_t^{(i)})_t$) to be added or removed.*

*Let $S_v = \sum_{t \in I_v} u_t$ be the aggregate over the interval $I_v$ for each node $v$ of the binary tree on $[T]$, and let the tree mechanism release $Z_v = S_v + \eta_v$ with independent $\eta_v \sim \text{Lap}(\lambda)$ for each $v$.*

*Then the $\ell_1$ sensitivity of the vector $(S_v)_v$ under this unit-level adjacency is at most $L \log_2 T$. In particular, if*

$$\lambda \;\geq\; \frac{L \log_2 T}{\varepsilon},$$

*then the tree mechanism is $\varepsilon$-differentially private with respect to unit-level adjacency.*

*Proof.* Fix a unit $i$, and consider two streams that differ only by the presence or absence of this unit's contributions $(u_t^{(i)})_t$. For each node $v$ in the binary tree, let $S_v^{(i)} = \sum_{t \in I_v} u_t^{(i)}$ be the contribution of unit $i$ to the interval sum at $v$. The change in $S_v$ under the addition or removal of unit $i$ is exactly $S_v^{(i)}$, so the $\ell_1$ sensitivity of the vector $(S_v)_v$ is

$$\sum_v |S_v^{(i)}|.$$

Each time step $t$ belongs to at most $\log_2 T$ node intervals $I_v$ in the binary tree, and $u_t^{(i)}$ contributes to $S_v^{(i)}$ precisely for those nodes. Therefore,

$$\sum_v \left| S_v^{(i)} \right| \;\leq\; \sum_{t=1}^{T} |u_t^{(i)}| \cdot \left( \#\{v : t \in I_v\} \right) \;\leq\; L \log_2 T.$$

Thus the $\ell_1$ sensitivity of $(S_v)_v$ under unit-level adjacency is at most $L \log_2 T$. By the standard Laplace mechanism analysis, adding independent $\mathrm{Lap}(\lambda)$ noise with $\lambda \geq L \log_2 T / \varepsilon$ to each coordinate $S_v$ yields $\varepsilon$-differential privacy for the vector $(Z_v)_v$. Finally, the prefix-sum outputs $(\tilde{S}_t)_t$ are deterministic functions of the noised node values $(Z_v)_v$, and hence inherit $\varepsilon$-DP by post-processing. $\qquad\square$

# H. Lemmas

**Lemma H.1** (Bound on the number of rate halving steps in Algorithm 2). *Let $T \in \mathbb{N}$ and let $(R_u)_{u=1}^{T}$ be i.i.d. random variables drawn from $\mathrm{Unif}[0, 1]$, representing the fixed sampling seeds of all units that ever appear in the stream. Let $R_{(m)}$ denote the $m$-th order statistic (the $m$-th smallest value) among $\{R_u\}_{u=1}^{T}$.*

*Fix a sample budget $k$ and define $m := k/2$. Then for any $\delta \in (0, 1)$, if*

$$k \;\geq\; 8 \log \frac{1}{\delta},$$

*it holds with probability at least $1 - \delta$ that*

$$R_{(k/2)} \;\leq\; \frac{k}{T}.$$

*Equivalently, with probability at least $1 - \delta$, for every integer*

$$j \;\geq\; \left\lceil \log_2 \left( \frac{T}{k} \right) \right\rceil,$$

*we have*

$$\#\{u : R_u \leq 2^{-j}\} \;\geq\; \frac{k}{2}.$$

*In particular, any algorithm that repeatedly halves its sampling rate $p = 2^{-j}$ and triggers a rate decrease whenever the maintained sample size exceeds $k$ will perform at most*

$$O\left( \log \frac{T}{k} \right)$$

*rate halving steps with probability at least $1 - \delta$.*

