# OpenReview forum: "Adaptively Robust Resettable Streaming"
_ICML.cc/2026/Conference — ICML 2026 regular_

### Official Review · Reviewer_SW79 · 2026-02-23

**Soundness:** 3
**Presentation:** 3
**Significance:** 3
**Originality:** 3
**Overall Recommendation:** 5
**Confidence:** 3

**Summary:**

This paper studies the problem in data streaming about handling with "resettable" data, where data entries can be updated or suddenly wiped back to zero. The problem is that current algorithms are sitting ducks for "adaptive attackers" who watch the algorithm’s outputs to figure out its internal randomness and then manipulate the stream to cause huge errors.

Technically, the authors solve this by borrowing concepts from differential privacy, especially some ideas from privacy under continous observation. From a technical perspective, the core strategy involves treating the sequence of updates to the sample size as a private stream and releasing noisy prefix sums through the tree structure. This DP layer effectively masks the specific membership of keys in the sample, ensuring that an attacker cannot learn enough about the internal state to launch targeted reset attacks. The proof leverages the generalization properties of DP to show that the internal randomness remains "protected" over the entire stream, allowing the algorithm to provide accurate "prefix-max" error guarantees for a variety of sublinear statistics, including cardinality and Bernstein functions

**Compliance With Llm Reviewing Policy:**

Affirmed.

**Final Justification:**

I found this paper to be good and keep my positve rating.

**Key Questions For Authors:**

I have no questions, congrats on the results.

**Limitations:**

yes

**Strengths And Weaknesses:**

This paper presents the first set of streaming algorithms that are adaptively robust in the "resettable" model, where data can be cleared or reset. Further, the design is not limited to simple counting; it supports a wide range of statistics including cardinality, sums, and complex Bernstein functions like frequency capping and entropy.

This paper is another evidence of the generalization/rubostness of differential privacy algorithms, where DP mechanisms are used to "hide" the algorithm's internal state from attackers without significantly sacrificing accuracy or performance. I really enjoy seeing papers that use tools from differential privacy or cryptography to solve algorithmic problems, though the results in this paper are not "super" surprising given existing papers like [HKM+20] or [BKM+22].

[HKM+20] Adversarially robust streaming algorithms via differential privacy. NeurIPS 2020

[NKM+22] Dynamic algorithms against an adaptive adversary: generic constructions and lower bounds. STOC 2022

---

> ### Author Rebuttal · Authors · 2026-03-30
>
> We are grateful for the very positive and thorough review. We appreciate that the reviewer recognizes the generality of our framework for a wide range of statistical estimation problems, and we are encouraged to see that the reviewer views this work as a strong contribution to the line of work on adversarially robust streaming.

---

> > ### Author Rebuttal · Reviewer_SW79 · 2026-04-01
> >
> > Thank you for your response. I will keep my positive score as it is.

---

### Official Review · Reviewer_rTyh · 2026-03-13

**Soundness:** 3
**Presentation:** 3
**Significance:** 3
**Originality:** 3
**Overall Recommendation:** 5
**Confidence:** 3

**Summary:**

This paper studies the adaptive/robust version of the "resettable streaming model" in sketching and streaming algorithms:

Assume there exists a set of keys and values $(x, v_x)$, and at each step one of the two following actions are possible to adjust the value for a given key: either $v_x \gets v_x + \Delta$ for some nonnegative $\Delta$ or $v_x \gets 0$ and it gets reset.
The goal is to maintain some estimate of the aggregated value $F:= \sum_x f(v_x)$ for some $f$ of interest. In particular in this paper they consider three cases for $f$: 1. $f(v) = 1[v > 0]$, which corresponds to cardinality or $\ell_0$ norm, 2. $f(v) = v$, which corresponds to the sum or $\ell_1$ norm, 3. and in general Bernstein functions (which includes the first two cases). For these cases they give algorithms that are able to maintain good approximations of $F$ throughout updates. Specifically, they show that using sketches of size $\text{poly}(1/\epsilon, \log(T_{inc}/\delta))$ where $T_{inc}$ is the number of increments one can maintain approximates $\hat{F}$ such that with probability $1-\delta$ we have for all $t$, $$ |\hat{F}_t - F_t | \le \epsilon \max_{t'\le t} F_{t'}$$  when the queries are allowed to be adaptive to the outputted estimates $F_{t'}$. They also give justification as why this is necessary and as opposed to the resettable setting you need to compare against $ \max_{t'\le t} F_{t'}$ as opposed to just $F_t$.

Technically, they use differential privacy together with the binary tree mechanism from continual observation literature to robustify the sketch against adaptive queries.

**Compliance With Llm Reviewing Policy:**

Affirmed.

**Final Justification:**

This is a nice paper on an interesting and well-motivated problem. The results seem technically sound, and my main initial concern was how the work compares to prior uses of differential privacy to robustify sketches. The rebuttal clarified this well, particularly the non-black-box nature of the approach and the improved space bounds in the resettable setting. This made the paper’s originality and significance clearer to me. Overall, I am updating my score and recommending acceptance.

**Key Questions For Authors:**

The two first cases $\ell_0$ and $\ell_1$ both seem well motivated, however the Bernstein statistic setting seems a bit niche to me. What is the motivation to consider this setting?

It's currently unclear to me how standard / novel the techniques are. I know differential privacy has been used to help robustify sketches in sketching / streaming settings in the past. Is the main technical novelty of this work in using the binary tree mechanism? Has btm been used in the context of streaming before?

**Limitations:**

yes

**Strengths And Weaknesses:**

The proofs and techniques all make sense and seem sound and reasonable at a high level. I don't see any immediate issues.

The presentation is also clear, they have also covered the literature well in the related work discussion. That being said, I would have liked to see some discussion of the techniques to show accuracy for the algorithms in the main body.

The setting seems natural, simple and theoretically interesting. They have given the first algorithm for it with a sublinear sketch size. They have also motivated some potential applications of this such as active entities or resource usage and removing people's data upon request (and updating aggregated statistics based on their data).

In terms of originality and novelty of the techniques, my understanding is that robustifying sketches have been previously used in streaming algorithms. It seems to me the main novelty of this work lies in using the binary tree mechanism. See questions below for more details.

Overall I think this is a good paper and I lean towards accepting the paper.

---

> ### Author Rebuttal · Authors · 2026-03-30
>
> We are grateful for the positive and detailed review. We appreciate that the reviewer finds the adaptive streaming model with resettable inputs to be a theoretically interesting and natural setting which is well-motivated by practical applications. We address specific questions below:
>
>
> 1. > The two first cases  and  both seem well motivated, however the Bernstein statistic setting seems a bit niche to me. What is the motivation to consider this setting?
>
>     The class of Bernstein functions includes several important statistics that are commonly studied in the streaming literature. For example, this includes $\ell_p$ norms for $p \in (0,1)$, which often appear as subroutines in empirical entropy estimation. Additionally, frequency cap functions have been extensively studied in the streaming model, first due to their widespread applications in advertising [Cohen 2015], as well as their theoretical application to designing algorithms for more complicated functions (for which capping functions form a ``basis'') [Cohen 2017], [Cohen-Geri 2019].
>
>
> 2. >   It's currently unclear to me how standard / novel the techniques are. I know differential privacy has been used to help robustify sketches in sketching / streaming settings in the past. Is the main technical novelty of this work in using the binary tree mechanism? Has btm been used in the context of streaming before?
>
>     Thank you for this question. It is true that differential privacy has been used in prior works to "robustify" sketches by protecting the internal randomness. Many of the approaches in previous works were largely ``black box'': in particular, given an oblivious (non-adaptive) streaming algorithm for a statistic $f$, the DP-based robustification framework instantiated several copies of the oblivious algorithm $\mathcal{A}$, and privately combined their individual estimates in a way that protected each algorithm's internal random bits. This black-box approach can provably only achieve space $\Theta(\sqrt{\lambda_\varepsilon(f)} \cdot \textrm{Space}(\mathcal{A}))$, where $\lambda_{\varepsilon}(f)$ is the maximum number of times that the value of statistic $f$ can change by a factor of $(1+\varepsilon)$ throughout the stream [HKMMS 2020]. In particular, this means that for *resettable* streams where the value of the statistic can change drastically due to resets at each time, previous approaches achieve $\tilde O(\sqrt{T})$ space for the statistics we consider.
>
>     On the other hand, our algorithm significantly departs from this common framework by carefully using differential privacy in a non-black box way, and achieves *exponentially* smaller $\textrm{polylog}(T)$ space: specifically, our approach involves carefully identifying the sensitive privacy units of particular oblivious streaming algorithms, and  protecting those units by using the binary tree mechanism. For instance, for $\ell_1$ estimation, we crucially relied on a clever decomposition of the classical estimator into a low-sensitivity private component (which we protect using DP) and a high sensitivity public component (which can be known to the adversary), thus enabling us to apply the binary tree mechanism on the protected component while maintaining sufficiently low error. In fact, note that without our careful decomposition, we would *not* be able to achieve prefix-max error in sublinear space. Moreover, the privacy units (i.e. sample membership for $\ell_0$, and thresholds $R_{x,j}$ in the case of $\ell_1$) may impact multiple updates in the stream and thus we proved a generalized privacy guarantee of the binary tree mechanism to account for this. To the best of our knowledge, this is the first time that an exponential improvement in space complexity has been achieved via techniques from differential privacy.
>
>
> References:
> 1. [Cohen 2015]: Stream Sampling for Frequency Cap Statistics (KDD 2015).
> 2.  [Cohen 2017]: HyperLogLog Hyper Extended: Sketches for Concave Sublinear Frequency Statistics (KDD 2017).
> 3. [Cohen-Geri 2019]: Sampling Sketches for Concave Sublinear Functions of Frequencies (NeurIPS 2019).
> 4. [HKMMS 2020]: Adversarially Robust Streaming Algorithms via Diﬀerential Privacy (NeurIPS 2020 and Journal of the ACM 2022).

---

> > ### Author Rebuttal · Reviewer_rTyh · 2026-04-04
> >
> > Thank you for the detailed rebuttal. The clarifications were very helpful, particularly on the novelty of the techniques. I appreciate the authors’ response, and I have updated my score.

---

> > > ### Author Response · Authors · 2026-04-04
> > >
> > > Thank you!

---

### Official Review · Reviewer_JdM6 · 2026-03-15

**Soundness:** 3
**Presentation:** 2
**Significance:** 3
**Originality:** 3
**Overall Recommendation:** 4
**Confidence:** 3

**Summary:**

The paper explores the setting of  designing space-efficient streaming sketches in the resettable streaming setting -- i.e. the stream is chosen adaptively and supports reset operations. Prior work with polylog space don't consider adaptive inputs. The paper shows that existing sampling-based sketches are vulnerable to adaptive attacks. The paper tackles this challenge and propose an adaptively robust sketch for resettable streams that require polylog space. The solution supports sublinear statistics like cardinality, sum and Bernstein statistics and is measured via prefix-max error.

**Compliance With Llm Reviewing Policy:**

Affirmed.

**Key Questions For Authors:**

The presented prefix-max error is weaker than relative error. Could the authors provide a discussion on this - in which practical scenarios is this guarantee is sufficient and what are the limitations of this error guarantee?

The proposed solution currently works for sublinear statistics like  cardinality, sum, and Bernstein functions. Is there a way to extend them to super linear statistics such as \ell_2?

**Limitations:**

no the paper does not include a discussion on the limitations

**Strengths And Weaknesses:**

The paper's core technical contribution are sketches for the resettable streaming model that are robust to adaptive inputs. In particular, the solution proposed requires polylog space. I found he central idea of the paper -- using differential privacy to hide the internal randomness of the sketch to be quite elegant. The application of the binary tree mechanism to streaming sketches is also quite novel.

One note about the presentation of the paper - the text is quite dense and a bit hard to read, the presentation can be improved upon.

A few things that the paper can do some clarification on:
Is the restriction to sub-linear statistics somehow inherent here? What would need to be done to extend and generalize the techniques to superlinear statistics such as \ell_2?

Also, I think adding a discussion on the limitations of prefix-max error would be helpful since it is weaker than the notion of relative error. In particular, some practical use cases where prefix-max works fine and a discussion where we should perhaps desire relative error.

---

> ### Author Rebuttal · Authors · 2026-03-30
>
> We are very grateful for the positive and detailed review. We appreciate that the reviewer recognized the elegance and novelty of our algorithms for the adaptive resettable streaming setting. We will make sure to improve the readability of the paper in the final version. We further address questions raised below:
>
> 1. > The presented prefix-max error is weaker than relative error. Could the authors provide a discussion on this - in which practical scenarios is this guarantee is sufficient and what are the limitations of this error guarantee?
>
> First, we emphasize that achieving $(1+\varepsilon)$ relative error in the resettable streaming setting would require the algorithm to store $\Omega(n)$ bits of memory, where $n$ is the size of the underlying universe. In particular, this means that it is strictly necessary to relax the error guarantee in order to achieve sublinear space, even without considering robustness to adaptive inputs. Note that the prefix-max error guarantee is standard for resettable streaming algorithms ([PCVM '24], [LNSW '26]). This lower bound follows by a reduction from the set-disjointness problem in one-way communication complexity: in the classical set-disjointness problem, Alice is given a set $A \subset [n]$ and Bob is given $B \subset [n]$, and after Alice sends a (one-way) message to Bob, Bob should decide whether $A \cap B = \emptyset$. It is known that this problem requires Alice to send $\Omega(n)$ bits of information if Bob is to correctly decide whether the sets are disjoint with constant success probability. Now, note that if there exists a  $(1+\varepsilon)$-relative error resettable streaming algorithm $\mathcal{S}$ for cardinality estimation, then Alice could simply run $\mathcal{S}$ on her input and send the current state $\mathcal{S}(A)$ to Bob. Then, Bob can continue running the streaming algorithm on the update sequence which *resets* all elements in $U \setminus B$. As a result, Bob obtains a sketch $\mathcal{S}(A \cap B)$. If $\mathcal{S}$ outputs a $(1+\varepsilon)$ approximation to the number of distinct elements in $A \cap B$, Bob will be able to *exactly* decide the set-disjointness problem, so this means that the $\Omega(n)$ bit lower bound for set-disjointness transfers to the resettable streaming setting.
>
> We note that the prefix-max error guarantee is sufficient for many applications where the value of the statistic always stays within a constant factor of the prefix-max, e.g. if the statistic is always in some range $[f_{\ell}, f_{h}]$ for $f_h/f_{\ell} = O(1)$, then $\varepsilon \cdot \max_{t' \leq t} F_{t'}$ corresponds to multiplicative error $O(\varepsilon) \cdot F_t$ (in fact, the gap $f_h/f_{\ell}$ can even be taken to be $O(\log n)$, by setting $\varepsilon' = \varepsilon/\log n$ we would still get multiplicative error). This error guarantee is arguably less satisfying when the statistic fluctuates significantly (i.e. increases/decreases by a polynomial $n^c$ factor) throughout the stream, but as explained above, we cannot hope to achieve standard multiplicative error in sublinear space.
>
> 2. > The proposed solution currently works for sublinear statistics like cardinality, sum, and Bernstein functions. Is there a way to extend them to super linear statistics such as $\ell_2$?
>
> The main obstacle in designing adaptively robust sketches for super-linear statistics (e.g. $\ell_p$ for $p > 1$) is that all known sampling sketches rely on persistent per-key randomness which is not ``refreshed'' throughout the stream ([MW 2010], [JW 2018], [XWZ 2025]). This inherently makes known approaches vulnerable to adaptive adversaries, and it is not clear how to appropriately define low-sensitivity privacy units for those sketches.
>
>
> References:
> 1. [PCVM 2024]: On the Feasibility of Forgetting in Data Streams (PODS 2024).
> 2. [LNSW 2026]: Unbiased Insights: Optimal Streaming Algorithms for $\ell_p$ Sampling, the Forget Model, and Beyond (PODS 2026).
> 3. [MW 2010]: $1$-pass Relative Error $L_p$ sampling with Applications (SODA 2010)
> 4. [JW 2018]: Perfect Lp Sampling in a Data Stream (FOCS 2018)
> 5.  [XWZ 2025]: Perfect Sampling in Turnstile Streams Beyond Small Moments (PODS 2025)

---

> > ### Author Rebuttal · Reviewer_JdM6 · 2026-04-04
> >
> > Thank you for the rebuttal. I will keep my positive score.

---

### Official Review · Reviewer_XP1N · 2026-03-19

**Soundness:** 3
**Presentation:** 3
**Significance:** 3
**Originality:** 3
**Overall Recommendation:** 5
**Confidence:** 1

**Summary:**

This paper considers the resettable streaming and shows that existing non-adaptive algorithms are vulnerable to adaptive adversarial attacks, such as the *sample-and-delete* attack. To overcome these vulnerabilities, the authors propose the first adaptively robust sketches for resettable streams. By leveraging the Binary Tree Mechanism from differential privacy, the proposed framework protects the internal randomness of the sketches. This approach successfully circumvents the space complexity limitations of previous defense methods and maintains polylogarithmic space complexity while supporting (sub)linear statistics including $L_0$, $L_1$ (sum), and Bernstein statistics

**Compliance With Llm Reviewing Policy:**

Affirmed.

**Final Justification:**

I keep my evaluation unchanged.

**Key Questions For Authors:**

No

**Limitations:**

Yes

**Strengths And Weaknesses:**

**Soundness**: This paper shows that authors are very familiar with existing literature in streaming algorithms. The theoretical results are presented with rigor and the use of DP look reasonable.

**Presentation**: This paper is clearly written and well-structured with a logical narrative flow that is easy to follow

**Significance**: This paper address a significant problem in the resettable streaming model where existing algorithms fails.  This theoretical framework is applicable to other domains. The scope of impact is well-justified.

**Originality**: The paper considers a very original problem in the resettable streaming problem. The use of Binary Tree Mechanism is very interesting.

---

> ### Author Rebuttal · Authors · 2026-03-30
>
> We thank the reviewer for the very positive review. We appreciate your recognition of the significance of our result, which gives the first known polylogarithmic space sketches in the adaptive resettable streaming model. We are pleased to hear that the writing was clear to follow and that the problem setting was well-motivated.

---

> > ### Author Rebuttal · Reviewer_XP1N · 2026-04-04
> >
> > My concerns have been adequately addressed

---

### Decision · Program_Chairs · 2026-04-30

**Decision:**

Accept (regular)

**Comment:**

The paper introduces a technically strong and timely framework for adaptively robust resettable streaming. Reviewers were generally positive after the rebuttal. The main contribution is the first polylog-space sketches for resettable streams that remain robust to adaptive adversaries, achieved via a non-black-box differential-privacy-based design and the binary tree mechanism. Reviewers found this approach elegant. The initial concerns were mainly on novelty relative to prior DP-based robustification work, the motivation for Bernstein statistics, and the strength of the prefix-max error guarantee, but the authors' rebuttal clarified why the new approach is genuinely different, why the broader statistic class is useful, and why stronger guarantees are not feasible in this setting.